# SDCBP/Syntenin-1 stabilizes BACH1 by disassembling the SCF^FBXO22–BACH1 complex in triple-negative breast cancer

Phi-Long Tran [1,5], Okhwa Kim [2,5], Cheol Hwangbo [3], Hyo-Jin Kim [3], Young-Myeong Kim [4] & Jeong-Hyung Lee [1,2 ✉]

## Abstract

**BACH1 is a redox-sensitive transcription factor facilitating tumor progression in triple-negative breast cancer (TNBC). However, the molecular mechanisms regulating BACH1 function in TNBC remain unclear. In this study, we demonstrate that SDCBP, a tandem-PDZ-domain protein, stabilizes BACH1 by disassembling the Skp1-Cullin1-FBXO22 (SCF^FBXO22)-BACH1 complex via a heme/heme-oxygenase-1-independent manner in TNBC cells. Our data revealed that SDCBP and BACH1 expression show a significant positive correlation in TNBC cells and TNBC patients tumor tissues. Mechanistically, SDCBP via its PDZ1 domain disassembles the SCF^FBXO22–BACH1 complex via its PDZ1 domain, thereby preventing BACH1 K48-linked polyubiquitination and proteasomal degradation. Knocking down SDCBP induces BACH1 degradation and downregulates expressions of BACH1-induced metastatic genes, thereby suppressing tumor progression in mice bearing TNBC tumors. Moreover, depleting SDCBP leads to upregulation of BACH1-repressed electron transport chain (ETC) genes, such as *NDUFA4* and *COX6B2*, and increases mitochondrial activity, enhancing anti-tumor efficacy of metformin against TNBC both in vitro and in vivo. These data demonstrate a novel alternative mechanism for BACH1 stabilization mediated by SDCBP, implicating the SDCBP-BACH1 axis as a potential target for enhancing ETC inhibitor efficacy in TNBC combinational therapy.**

**Keywords** BACH1; Metformin; SDCBP; Triple-negative Breast Cancer; Ubiquitin E3 Ligase SCF^FBXO22
**Subject Categories** Cancer; Signal Transduction

## Introduction

Breast cancer, a genetically and clinically heterogeneous disease, is an abundant malignancy in women and is the most diagnosed cancer worldwide (Sung et al, 2021). Approximately 15–20% of all patients diagnosed with breast cancer have triple-negative breast cancer (TNBC), which is the most aggressive subtype with a high level of drug resistance, metastatic probability, and poor prognosis (Garrido-Castro et al, 2019; Yin et al, 2020). Lacking estrogen receptor (ER), progesterone receptor (PR), and human epidermal growth factor-2 (HER2) expression allows TNBC to be highly resistant to endocrine and anti-HER2 targeted therapies (Yin et al, 2020). Consequently, effective curable options for TNBC are currently limited (Sung et al, 2021; Garrido-Castro et al, 2019; Yin et al, 2020).

BTB domain and CNC homology 1 (BACH1), a member of the Cap 'n' Collar and basic region leucine zipper family of transcription factors, plays several key roles in the oxidative stress response, heme homeostasis, senescence, tumor progression, and metastasis (Zhang et al, 2018; Ou et al, 2019; Sato et al, 2020). BACH1 functions as both transcriptional activator and transcriptional repressor of numerous genes in different cellular processes by binding to DNA elements known as MAF recognition motifs (MAREs) (Zhang et al, 2018; Amoutzias et al, 2007). BACH1 is involved in promoting metastasis by upregulating pro-metastatic genes such as *HK2*, *GAPDH*, *CXCR4*, *MMP1*, *MMP13*, and *VEGF*, and promotes the metastasis of cancer cells in vitro and in vivo (Liang et al, 2012; Lignitto et al, 2019; Wiel et al, 2019). BACH1 is highly expressed and is responsible for metastasis in several cancers, including TNBC, lung cancer, and pancreatic cancer (Sato et al, 2020; Lignitto et al, 2019; Wiel et al, 2019; Lee et al, 2013; Yun et al, 2011; Lee et al, 2019). In TNBC, BACH1 supports anaerobic glycolysis by repressing the transcription of genes involved in mitochondrial oxidative phosphorylation, including *COX15* and *UQCRC1*, and by positively regulating the transcription of *PDK* genes that inhibit the activity of the pyruvate dehydrogenase complex (Lee et al, 2019).

[1]Department of Biochemistry, College of Natural Sciences, Kangwon National University, Chuncheon-si, Gangwon-do 24341, Republic of Korea. [2]Kangwon Institute of Inclusive Technology, Kangwon National University, Chuncheon-si, Gangwon-do 24341, Republic of Korea. [3]Division of Applied Life Science (BK21 Four), Division of Life Science, College of Natural Sciences, Gyeongsang National University, Jinju-si, Gyeongsangnam-do 52828, Republic of Korea. [4]Department of Molecular and Cellular Biochemistry, School of Medicine, Kangwon National University, Chuncheon-si, Gangwon-do 24341, Republic of Korea. [5]These authors contributed equally: Phi-Long Tran, Okhwa Kim. ✉E-mail: jhlee36@kangwon.ac.kr

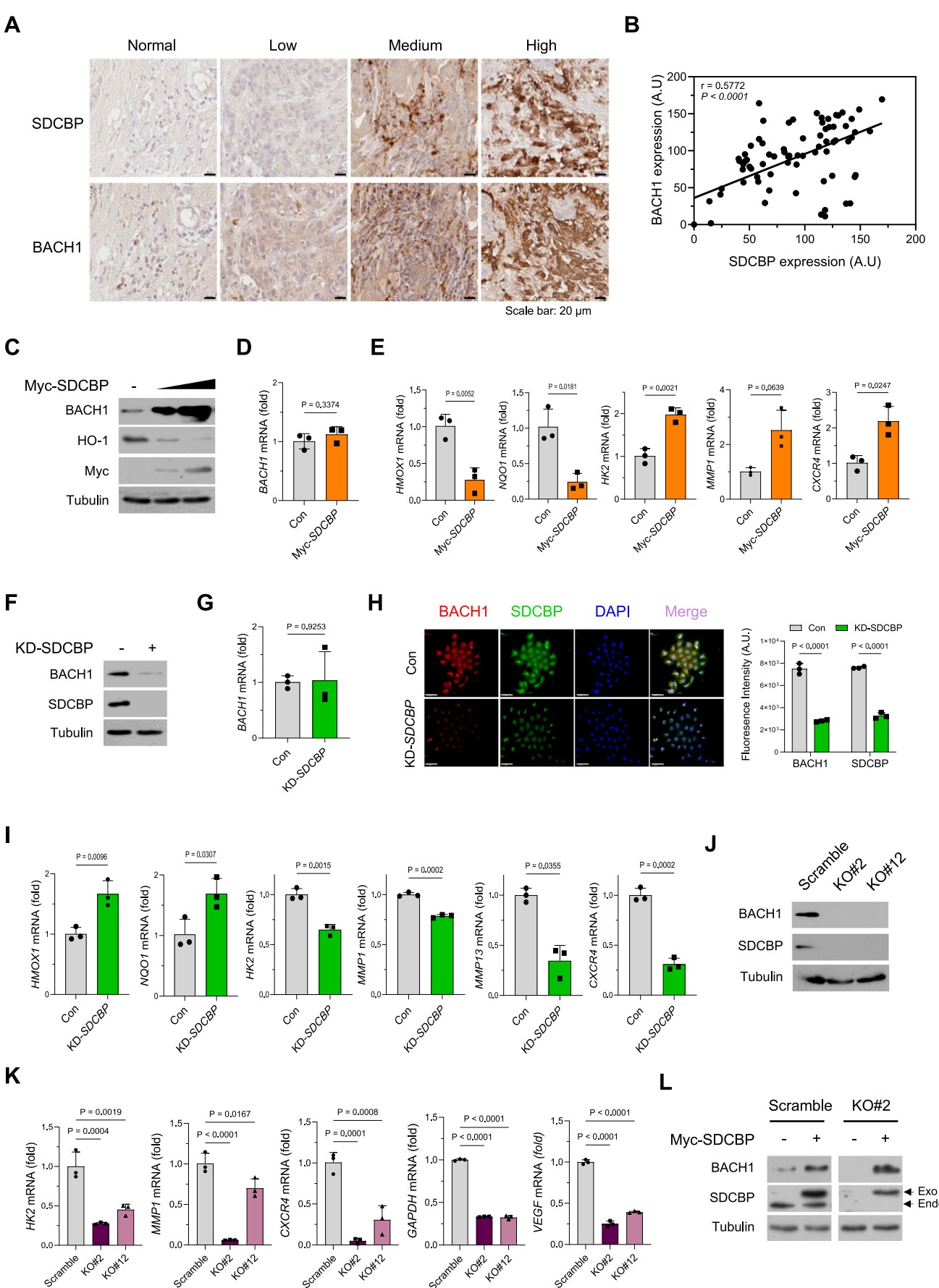

**Figure 1. SDCBP and BACH1 protein are co-expressed and SDCBP increases BACH1 protein levels in TNBC cells.**

(A) Immunohistochemistry staining against the SDCBP and BACH1 protein in human TNBC-derived tissue microarray sections ($n = 78$). Representative images showing the co-expression of SDCBP and BACH1 in the same section. Normal breast cancer tissues were considered as the negative control. Scale bar = 20 μm. (B) Pearson correlation coefficient ($r = 0.5772$, $P < 0.0001$) between SDCBP and BACH1 expression in (A). (C) Western blot showing BACH1 and HO-1 protein expression in Hs578T cells transfected with control vector or Myc-SDCBP-expressing vector. (D) Real-time qPCR showing *BACH1* mRNA expression in Hs578T cells transfected with a control vector or a Myc-SDCBP-expressing vector ($n = 3$). (E) Real-time qPCR showing the mRNA expression of BACH1-regulated antioxidant genes (*HMOX1* and *NQO1*) and BACH1-regulated metastatic genes (*HK2*, *MMP1*, and *CXCR4*) in Hs578T cells transfected with control vector or Myc-SDCBP-expressing vector ($n = 3$). (F) Western blot showing BACH1 protein expression in MDA-MB-231 infected with lentiviral scramble or SDCBP shRNA. (G) Real-time qPCR showing *BACH1* mRNA expression in MDA-MB-231 infected with lentiviral scramble or SDCBP shRNA ($n = 3$). (H) Left, Representative images of immunofluorescence staining to visualize SDCBP (*green color*) and BACH1 (*red color*) expression in MDA-MB-231 cells transfected with a scramble siRNA or SDCBP siRNA. DAPI (*blue color*) was used to stain the nucleus ($n = 3$); Scale bar = 50 μm. Right, fluorescence levels of SDCBP and BACH1 were quantified based on their spectral densities. (I) Real-time qPCR showing the mRNA expression of BACH1-regulated antioxidant genes (*HMOX1* and *NQO1*) and BACH1-regulated metastatic genes (*HK2*, *MMP1*, *MMP13*, and *CXCR4*) in MDA-MB-231 cells transfected with scramble siRNA or SDCBP siRNA ($n = 3$). (J) Western blot showing SDCBP and BACH1 protein expression in scramble control and two SDCBP-KO MDA-MB-231 clones (KO#2, KO#12) generated using CRISPR-Cas9 system ($n = 3$). (K) Real-time qPCR showing the mRNA expression of BACH1-regulated metastatic genes (*HK2*, *MMP1*, *CXCR4*, *GAPDH*, and *VEGF*) in scramble control and SDCBP-KO MDA-MB-231 cells. (L) The reconstitution of SDCBP recovers BACH1 protein expression in SDCBP-KO MDA-MB-231 cells. Western blot showing BACH1 protein expression in scramble control and SDCBP-KO MDA-MB-231 cells transfected with control vector or Myc-SDCBP-expressing vector. The arrows indicate the endogenous (Endo) and exogenous (Exo) SDCBP. Data are expressed as the mean ± SEM and analyzed using two-tailed Student's *t* test with Welch's correction (D, E, G, I), two-way ANOVA (H), or one-way ANOVA (K). *P* values less than 0.05 were considered statistically significant. All experiments were repeated at least three times unless otherwise indicated. Source data are available online for this figure.

BACH1 stability is critically regulated by free heme, which binds to Cys-Pro motifs in BACH1, inducing nuclear export, polyubiquitination, and subsequent degradation (Sun et al, 2002; Suzuki et al, 2004). Three E3 ubiquitin ligases are thought to be responsible for this polyubiquitination-mediated degradation. HOIL1 (heme-oxidized IRP2 ubiquitin ligase-1) encoded by *RBCK1* is responsible for heme-bound BACH1, while the SKP1-CUL1-F box protein (SCF) E3 complexes, SCF^FBXO22 and SCF^FBXL17, are involved in this regulatory process (Zenke-Kawasaki et al, 2007; Lignitto et al, 2019). The F-box proteins, which determine substrate specificity, contain at least one F-box domain and are classified into three subclasses, based on the presence of specific substrate recognition domains (Wang et al, 2014; Frescas and Pagano, 2008). Like most FBXO proteins, FBXO22 contains the F-box domain in the N-terminus and protein–protein interaction domains for substrate recognition in the C-terminus and FBXO22 has been shown to regulate tumor progression and metastasis by targeting key regulators like BACH1 (Lignitto et al, 2019). However, the detailed mechanisms by which SCF^FBXO22 recognizes and degrades BACH1 in the context of TNBC remain unclear.

SDCBP, encoded by *SDCBP1*, also known as syntenin-1, is a PDZ domain-containing protein highly expressed in numerous cancers, including TNBC, and correlates with poor prognosis (Liu et al, 2018). SDCBP plays crucial roles in regulating various processes during tumor progression, including proliferation, extracellular trafficking, immunosuppression, angiogenesis, and tumor metastasis (Tae et al, 2017; Kim et al, 2022; Pradhan et al, 2020). SDCBP is also suggested as a tumor biomarker as well as a promising therapeutic target in breast cancer treatments (Liu et al, 2018; Lee et al, 2023). However, the function of SDCBP in TNBC tumor progression and in BACH1 stability remains elusive.

Given that the mechanisms regulating BACH1 stability and degradation as well as those by which FBXO22 recognizes BACH1 require clarification, the current study was aimed at identifying a novel molecular mechanism for SDCBP-dependent BACH1 stabilization in TNBC cells. We found that SDCBP stabilizes BACH1 by inhibiting SCF^FBXO22-mediated K48-linked polyubiquitination and degradation in an heme/HO-1-independent manner. Furthermore, the role of the SDCBP–BACH1 axis in

tumor progression and metastasis, both in vitro and in vivo, was assessed. This study could provide valuable insights for disassembling the SCF^FBXO22-BACH1 complex and targeting the SDCBP-BACH1 axis as a treatment strategy for TNBC.

## Results

### SDCBP and BACH1 protein are co-expressed and SDCBP increases BACH1 protein levels in TNBC cells

We first examined the relationship between SDCBP and BACH1. Analyses of TNBC datasets from The Cancer Genome Atlas (TCGA) suggested a positive correlation between *SDCBP* and *BACH1* mRNA expressions (Fig. EV1A,B). We next explored the association between SDCBP and BACH1 protein levels in tissues of TNBC patients. Immunohistochemistry staining of human TNBC tissue microarray indicated detectable expressions of SDCBP and BACH1 in the tissues of patients with TNBC; while there was little or no staining expression detected in normal breast tissues (Fig. 1A). In addition, we found a significant positive correlation between SDCBP and BACH1 expression ($P < 0.0001$; Pearson correlation coefficient = 0.5772) in 78 tumor tissue samples from patients with TNBC (Fig. 1A,B), and this Pearson correlation coefficient was more reliable than that observed in *SDCBP* and *BACH1* mRNA expression (Fig. EV1A,B), suggesting a strong positive correlation between SDCBP and BACH1 protein expression in TNBC tissues. We then investigated SDCBP and BACH1 expression levels in several human breast cancer cell lines. Western blot analyses showed high SDCBP and BACH1 protein expressions in TNBC cell lines (e.g., MDA-MB-231, MDA-MB-468, and Hs578T), while lower levels were found in luminal A breast cancer cell lines (e.g., MCF-7 and T47D) (Fig. EV1C,D). Notably, MDA-MB-231 cells exhibited the highest expression levels of BACH1 and SDCBP but the lowest expression level for HO-1 (encoded by *HMOX1*), a gene repressed by BACH1 (Fig. EV1C). Real-time quantitative polymerase chain reaction (qPCR) analyses revealed that the mRNA expression levels of *SDCBP*, but not those of *BACH1*, correlated with the protein expression levels in the breast

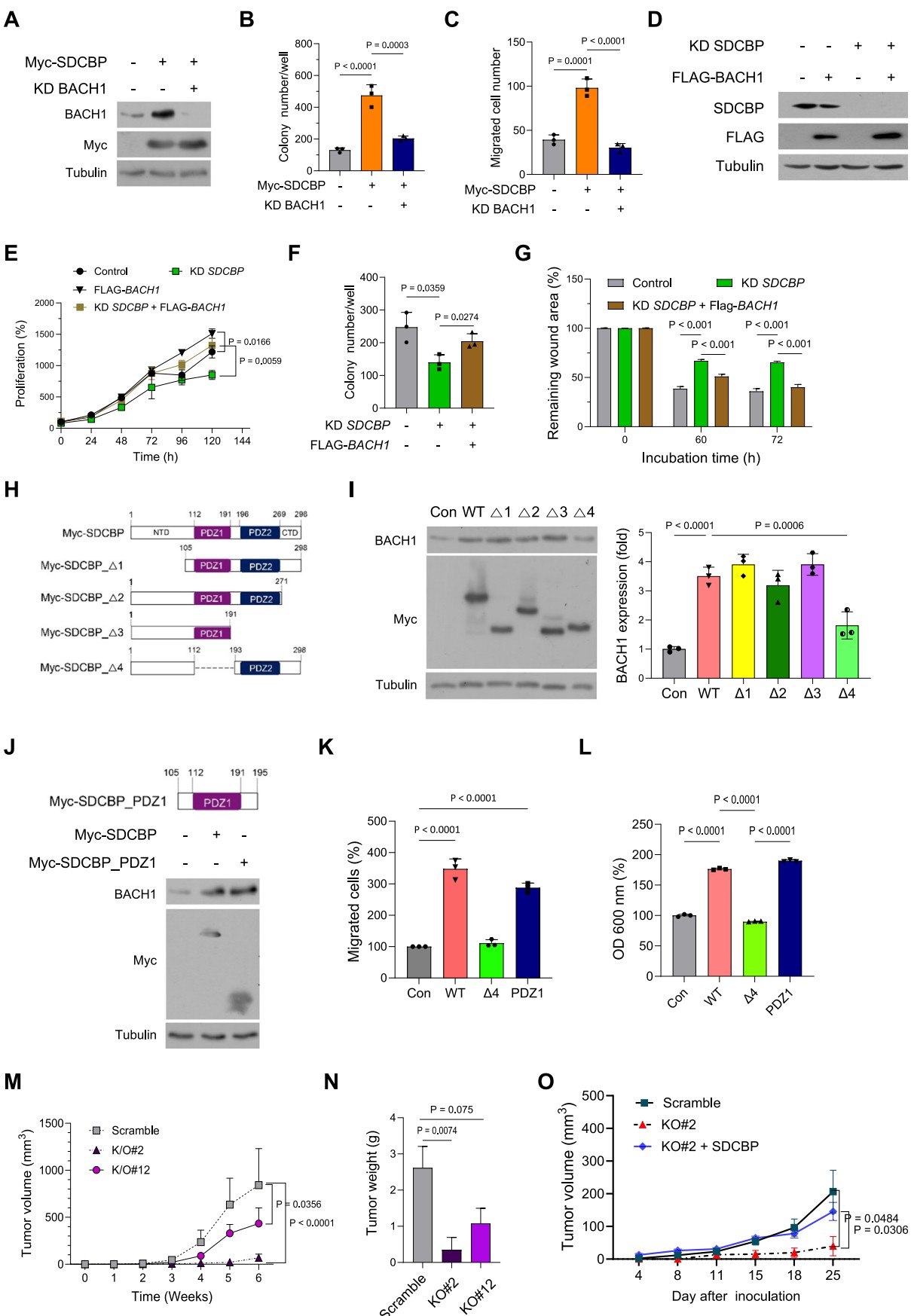

**Figure 2. SDCBP promotes tumor progression by upregulating BACH1 protein via its PDZ1 domain in TNBC cells.**

(A) Western blot showing BACH1 protein expression in Hs578T cells transfected with Myc-SDCBP-expressing vector with or without BACH1 siRNA. (B) Colony formation of Hs578T cells transfected with Myc-SDCBP-expressing vector with or without BACH1 siRNA ($n = 3$); See also Fig. EV2A. (C) Migration of Hs578T cells transfected with Myc-SDCBP-expressing vector with or without BACH1 siRNA ($n = 3$); See also Fig. EV2B. (D) Western blot showing SDCBP and Flag-BACH1 protein expression in MDA-MB-231 cells transfected with SDCBP siRNA with or without a Flag-BACH1-expressing vector. (E) Cell proliferation of MDA-MB-231 cells transfected with SDCBP siRNA with or without Flag-BACH1-expressing vector ($n = 3$). (F) Colony formation of MDA-MB-231 cells transfected with SDCBP siRNA with or without Flag-BACH1-expressing vector ($n = 3$); See also Fig. EV2C. (G) Wound closure of scratched MDA-MB-231 cells transfected with SDCBP siRNA with or without a Flag-BACH1-expressing vector ($n = 3$); See also Fig. EV2D. (H) Schematic of various SDCBP mutant constructs generated using the Myc-SDCBP plasmid. (I) Left, western blot showing BACH1 protein expression in Hs578T cells transfected with the indicted SDCBP constructs. Right, quantification of BACH1 levels using densitometry ($n = 3$). (J) Top, schematic of the PDZ1 construct. Bottom, western blot showing BACH1 protein expression in Hs578T cells transfected with Myc-SDCBP or Myc-SDCBP_PDZ1 plasmid. (K) Migration of Hs578T cells transfected with Myc-SDCBP, Myc-SDCBP_Δ4, or Myc-SDCBP_PDZ1 plasmid ($n = 3$); See also Fig. EV2H. (L) Colony formation of Hs578T cells transfected with Myc-SDCBP, Myc-SDCBP_Δ4, or Myc-SDCBP_PDZ1 plasmid ($n = 3$); See also Fig. EV2I. (M) Tumor volumes from athymic BALB/c nude mice 6 weeks after mammary fat-pad injection of the scramble control or SDCBP-KO MDA-MB-231 cells ($1 \times 10^5$ cells/mouse; $n = 5$ or 7 mice/group). (N) Tumor weights in Fig. 2M ($n = 7$ mice/group); See also Fig. EV3D. (O) Tumor volumes from athymic BALB/c nude mice 25 days after mammary fat-pad injection of the scramble control, SDCBP-KO MDA-MB-231 cells, and SDCBP-KO MDA-MB-231 cells stably transfected with Flag-SDCBP ($1 \times 10^5$ cells/mouse; $n = 5$ mice/group); See also Fig. EV3A–C,E. Data are expressed as the mean ± SEM and analyzed using one-way ANOVA (B, C, I, K, L, N), two-tailed Student's $t$ test (E, F, O), or two-way ANOVA (G, M). $P$ values less than 0.05 were considered statistically significant. All experiments were repeated at least three times unless otherwise indicated. Source data are available online for this figure.

cancer cell lines (Fig. EV1D,E). These results indicated that SDCBP and BACH1 proteins are co-expressed in TNBC cells.

Next, we examined the impacts of SDCBP on BACH1 protein expression in TNBC cells. Overexpression of SDCBP in Hs578T cells increased BACH1 protein expression, while no changes were observed in BACH1 mRNA levels (Fig. 1C,D). Moreover, SDCBP overexpression downregulated antioxidant genes suppressed by BACH1, including *HMOX1* and *NQO1*, while simultaneously upregulating pro-metastatic genes controlled by BACH1, such as *HK2, MMP1,* and *CXCR4* (Fig. 1E). In contrast, the knockdown (KD) of SDCBP by shRNA or siRNA in MDA-MB-231 cells decreased BACH1 protein expression but did not affect BACH1 mRNA levels (Fig. 1F–H). Consistently, SDCBP KD upregulated expressions of *HMOX1* and *NQO1* while downregulating expressions of the BACH1-regulated pro-metastatic gene (Fig. 1I). Notably, BACH1 KD did not alter the SDCBP expression but did induce HO-1 expression in MDA-MB-231 cells (Fig. EV1F), suggesting that SDCBP is not a direct target of BACH1. To validate these findings, we utilized CRISPR/Cas9 technology to abrogate SDCBP expression in MDA-MB-231 and generated two SDCBP knockout (KO) clones (KO#2 and KO#12) for further studies (Fig. EV1G). Similarly, SDCBP-KO clones (KO#2 and KO#12) decreased expressions of BACH1 protein as well as BACH1-regulated pro-metastatic genes, while no changes in BACH1 mRNA levels were observed (Figs. 1J,K and EV1H,I). SDCBP KD in 4T1 murine TNBC cells consistently decreased the expression of the BACH1 protein, while increasing expressions of BACH1-regulated antioxidant genes, including *HMOX1, NQO1,* and *GCLC* (Fig. EV1J,K). Intriguingly, reconstitution of SDCBP in KO#2 cells restored BACH1 protein (Fig. 1L), indicating that SDCBP positively regulates BACH1 expression.

## SDCBP promotes tumor progression by upregulating BACH1 protein via its PDZ1 domain in TNBC cells

We further investigated the role of BACH1 in SDCBP-induced proliferation, colony formation, and migration in TNBC cells. SDCBP overexpression increased the colony formation and migration in Hs578T cells; however, BACH1 KD reversed these SDCBP-induced effects (Figs. 2A–C and EV2A,B). In contrast,

SDCBP KD inhibited proliferation, colony formation, and migration, whereas BACH1 overexpression restored tumor progression in MDA-MB-231 cells (Figs. 2D–G and EV2C,D). Consistently, BACH1 KD reduced proliferation, colony formation, and migration in MDA-MB-231 cells (Fig. EV2E–G).

To identify the SDCBP domain responsible for BACH1 expression, a series of SDCBP mutants was constructed (Fig. 2H), and their impacts on BACH1 expression in Hs578T cells were examined. Overexpression of these SDCBP mutants increased BACH1 expression to a similar extent as SDCBP did, except for the SDCBP mutant (SDCBP_Δ4) with a deletion in the PDZ1 domain (Fig. 2I). Moreover, overexpression of the SDCBP_PDZ1 domain increased BACH1 expression (Fig. 2J). Consistent with impacts of BACH1 expression, overexpression of the PDZ1 domain alone increased colony formation, and migration of Hs578T cells; however, SDCBP_Δ4 eliminated these effects (Figs. 2K,L and EV2H,I). These results suggest that the PDZ1 domain of SDCBP is essential for SDCBP-induced BACH1 expression.

We next evaluated the impacts of SDCBP on tumor growth using the MDA-MB-231 xenograft model. SDCBP-KO MDA-MB-231 cells (KO#2 and KO#12) were injected into the mammary fat pad of athymic BALB/c nude mice. SDCBP KO resulted in a significant decrease in tumor growth compared to the control group (Fig. 2M,N). Re-expressing SDCBP in SDCBP-KO MDA-MB-231 cells (KO#2) restored tumor growth of MDA-MB-231 cells (Figs. 2O and EV3A–C). Immunohistochemical staining of the xenograft tumors further revealed that the SDCBP KO decreased the protein expression levels of BACH1 and the proliferation marker Ki67 (Fig. EV3D) and SDCBP re-expression reversed these effects (Fig. EV3E). These results suggested that SDCBP may promote tumor growth of TNBC cells by regulating BACH1 expression.

Kaplan–Meier analyses of the TCGA database showed that high *SDCBP* mRNA expression was associated with poor clinical outcomes of patients with TNBC (Fig. EV3F). Interestingly, high *BACH1* mRNA expression was not associated with poor survival rates of TNBC patients (Fig. EV3G). These findings suggested that BACH1 protein, rather than *BACH1* mRNA, plays more significant roles in tumor progression and aggressiveness of TNBC, and *BACH1* mRNA expression data may not provide a reliable prediction of survival outcomes in TNBC patients.

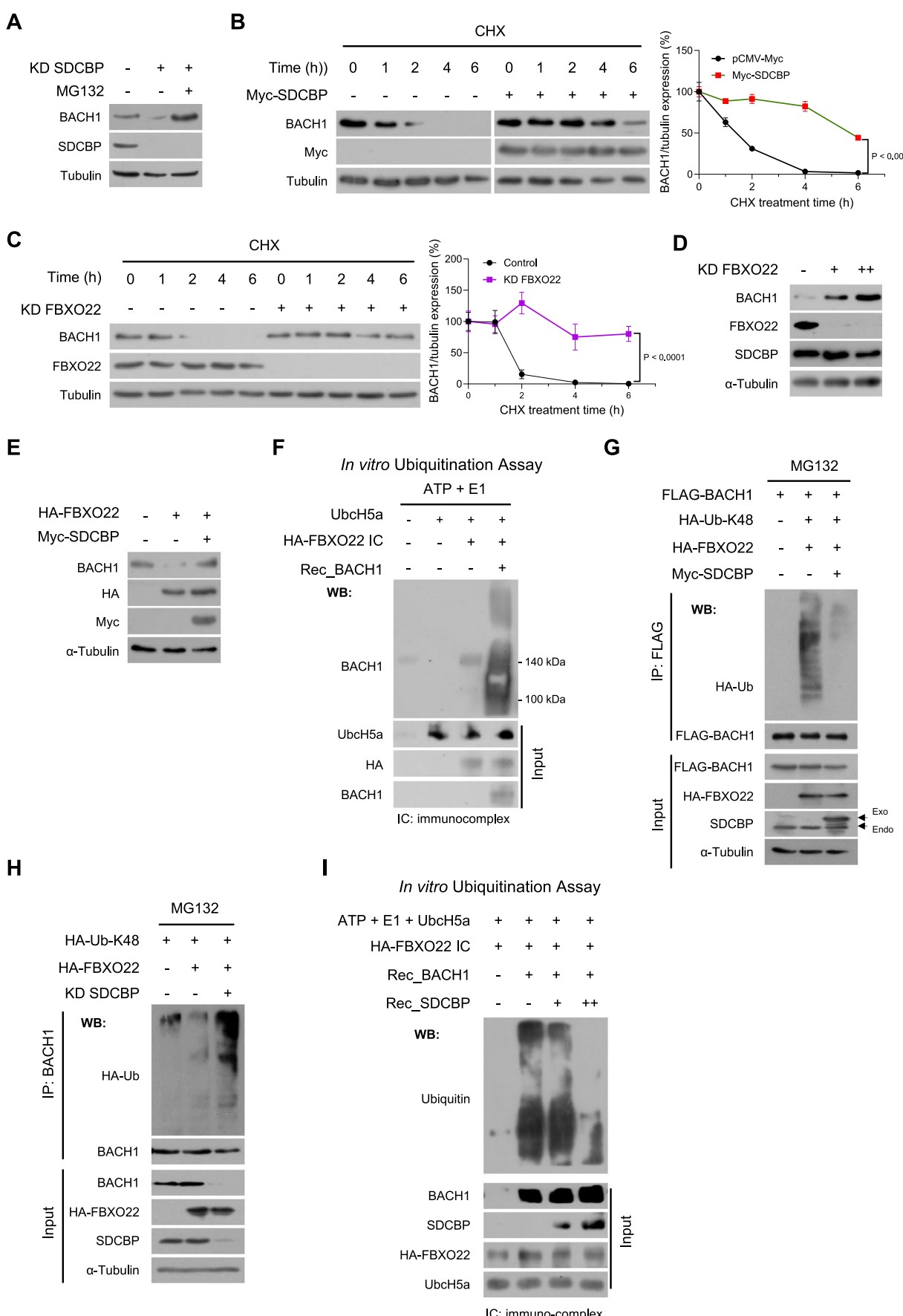

**Figure 3.  SDCBP stabilizes BACH1 by inhibiting SCF^FBXO22-mediated K48-linked polyubiquitination in a heme/HO-1-independent manner.**

(A) Western blot showing BACH1 protein expression in MDA-MB-231 cells transfected with SDCBP siRNA with or without MG132 proteasome inhibitor. (B) Left, western blot showing BACH1 protein expression in Hs578T cells transfected with control vector or Myc-SDCBP-expressing vector in the presence of CHX protein synthesis inhibitor at various time points. Right, quantification of BACH1 protein levels using densitometry ($n = 3$). (C) Left, western blot showing BACH1 protein expression in MDA-MB-231 cells transfected with scramble or FBXO22 siRNA in the presence of CHX protein synthesis inhibitor at various time points. Right, quantification of BACH1 protein levels using densitometry ($n = 3$). (D) Western blot showing BACH1, FBXO22, and SDCBP protein expression in MDA-MB-231 cells transfected with scramble or FBXO22 siRNA. (E) Western blot showing BACH1 protein expression in MDA-MB-231 cells transfected with HA-FBXO22-expressing vector with or without Myc-SDCBP-expressing vector. (F) In vitro ubiquitylation assay of the recombinant human BACH1 protein mediated by the FBXO22 complex. Active recombinant human UbcH5a protein was used as the E2 ubiquitin-conjugating enzyme for FBXO22 complex-mediated BACH1 degradative polyubiquitylation. (G) In vivo ubiquitylation assay showing the decrease in the K48-linked polyubiquitylation of BACH1 by SDCBP overexpression in HEK293 cells transfected with the indicated plasmids. (H) In vivo ubiquitylation assay showing the increase in the K48-linked polyubiquitylation of BACH1 by SDCBP KD in MDA-MB-231 cells transfected with the indicated plasmids. (I) In vitro ubiquitylation assay showing the inhibitory effect of SDCBP on the polyubiquitylation of BACH1 mediated by the FBXO22 complex. Active recombinant human protein UbcH5a and immunocomplex FBXO22 were added as described above. Recombinant human BACH1 and recombinant human SDCBP proteins were added at ratios of 1:1 (+) and 1:2 (++). Data are expressed as the mean ± SEM and analyzed using two-tailed Student's $t$ test with Welch's correction (B, C). P values less than 0.05 were considered statistically significant. All experiments were repeated at least three times unless otherwise indicated. Source data are available online for this figure.

## SDCBP stabilizes BACH1 by inhibiting SCF^FBXO22-mediated K48-linked polyubiquitination and degradation in an heme/HO-1-independent manner

We then investigated the underlying mechanisms by which SDCBP regulates BACH1 protein expression. Treatment with MG132 restored the BACH1 protein level in the SDCBP KD cells (Fig. 3A), indicating that SDCBP might regulate BACH1 stability. To confirm this result, we further investigated the BACH1 protein half-life using a protein synthesis inhibitor cycloheximide (CHX). Our results showed that SDCBP overexpression significantly extended the half-life of BACH1 protein in TNBC cells (Figs. 3B and EV4A).

We next questioned whether SDCBP affects intracellular heme levels to stabilize BACH1 protein in TNBC cells. Intriguingly, overexpression or KO of SDCBP did not alter intracellular heme levels (Fig. EV4B,C). Previous studies have reported that the NRF2/HO-1 pathway improve BACH1 stabilization in lung adenocarcinoma cells (Lignitto et al, 2019). We then investigated the impact of NRF2/HO-1 on BACH1 expression in TNBC cells. The ectopic expression of NRF2 (encoded by *NFE2L2*) or HO-1 (encoded by *HMOX1*) in MDA-MB-231 cells did not increase BACH1 expression but decreased it (Fig. EV4D,E). Furthermore, HO-1 KD in Hs578T cells also failed to decrease BACH1 expression (Fig. EV4F). Importantly, blocking the catalytic function of HO-1 by H25A mutation did not alter BACH1 expression (Fig. EV4G). These results demonstrated that SDCBP may stabilize BACH1 protein in a heme/HO-1-independent manner in TNBC cells.

BACH1 is the substrate of E3 ubiquitin HOIL1 (encoded by *RBCK1*) and E3 ubiquitin SKP1–CUL1–FBXO22 complex (SCF^FBXO22 complex) for K48-linked polyubiquitination and degradation (Lignitto et al, 2019; Zenke-Kawasaki et al, 2007). Our data showed that HOIL1 KD did not lead to an increase of BACH1 expression (Fig. EV4H). Therefore, we evaluated the impacts of SCF^FBXO22 on polyubiquitination and degradation of BACH1. At the basal level, MDA-MB-231, Hs578T, MCF-7, and T47D cells showed similar protein expression levels for FBXO22 (Fig. EV4I). Notably, FBXO22 KD prolonged the protein half-life of BACH1 and significantly increased BACH1 expression in MDA-MB-231 cells (Fig. 3C,D), indicating that FBXO22 may play a critical role in regulating BACH1 stability in TNBC cells.

It was questioned whether SDCBP inhibits the FBXO22-mediated polyubiquitination and degradation of BACH1 protein. SDCBP overexpression blocked the FBXO22-mediated

downregulation of BACH1 expression (Fig. 3E). Additionally, our data showed that FBXO22 interacted with BACH1 (Fig. EV4J) and induced BACH1 polyubiquitination in both HEK293 and MDA-MB-231 cells (Fig. EV4K,L), and in vitro without the addition of heme or hemin (Fig. 3F). SDCBP overexpression inhibited FBXO22-mediated K48-linked polyubiquitination of BACH1 in MDA-MB-231 and HEK293 cells (Figs. 3G and EV4M). In contrast, SDCBP KD increased FBXO22-mediated K48-linked polyubiquitination of BACH1 in MDA-MB-231 cells (Fig. 3H). In vitro ubiquitination assays also showed that FBXO22 ubiquitinated recombinant BACH1; however, the addition of recombinant SDCBP blocked SCF^FBXO22-mediated ubiquitination of BACH1 (Fig. 3I). These results suggested that SDCBP induces BACH1 stability by inhibiting SCF^FBXO22-mediated K48-linked polyubiquitination and subsequent degradation of BACH1 in TNBC cells.

## SDCBP PDZ1 domain plays an essential role in disruption of SCF^FBXO22–BACH1 complex

We continued to explore mechanisms by which SDCBP inhibits FBXO22-mediated K48-linked polyubiquitination and degradation of BACH1. Co-immunoprecipitation assays revealed that SDCBP was associated with FBXO22 (Fig. EV5A,B). Crosslink-immunoprecipitation assays further confirmed this association in Hs578T cells, demonstrating that FBXO22 was associated with SDCBP but not with SDCBP-Δ4 (Fig. 4A), indicating that the SDCBP PDZ1 domain may be responsible for the association with FBXO22. To identify the regions of FBXO22 which interact with SDCBP, FBXO22 mutants were constructed with deleted F-box (FBXO22_Δ1) or FIST-C (FBXO22_Δ2) domain, and co-immunoprecipitation assays were then performed (Fig. 4B). Deletion of the F-box or FIST-C domain in FBXO22 abrogated its association with SDCBP (Fig. 4C), indicating that both the F-box and FIST-C domains of FBXO22 may be necessary for its association with SDCBP. Notably, overexpression of SDCBP impaired the association of BACH1 with FBXO22 (Fig. EV5C). Further investigations were conducted to determine whether SDCBP regulates SCF^FBXO22–BACH1 complex formation using pull-down assays (Fig. EV5D). The results showed that SDCBP impaired the association of FBXO22 with SKP1 and strongly bound to SKP1 as a regulative adapter protein (Fig. EV5E; Appendix Fig. S1A). Moreover, SDCBP disrupted the formation of the SKP1-CUL1-FBXO22 complex, resulting in the formation of the

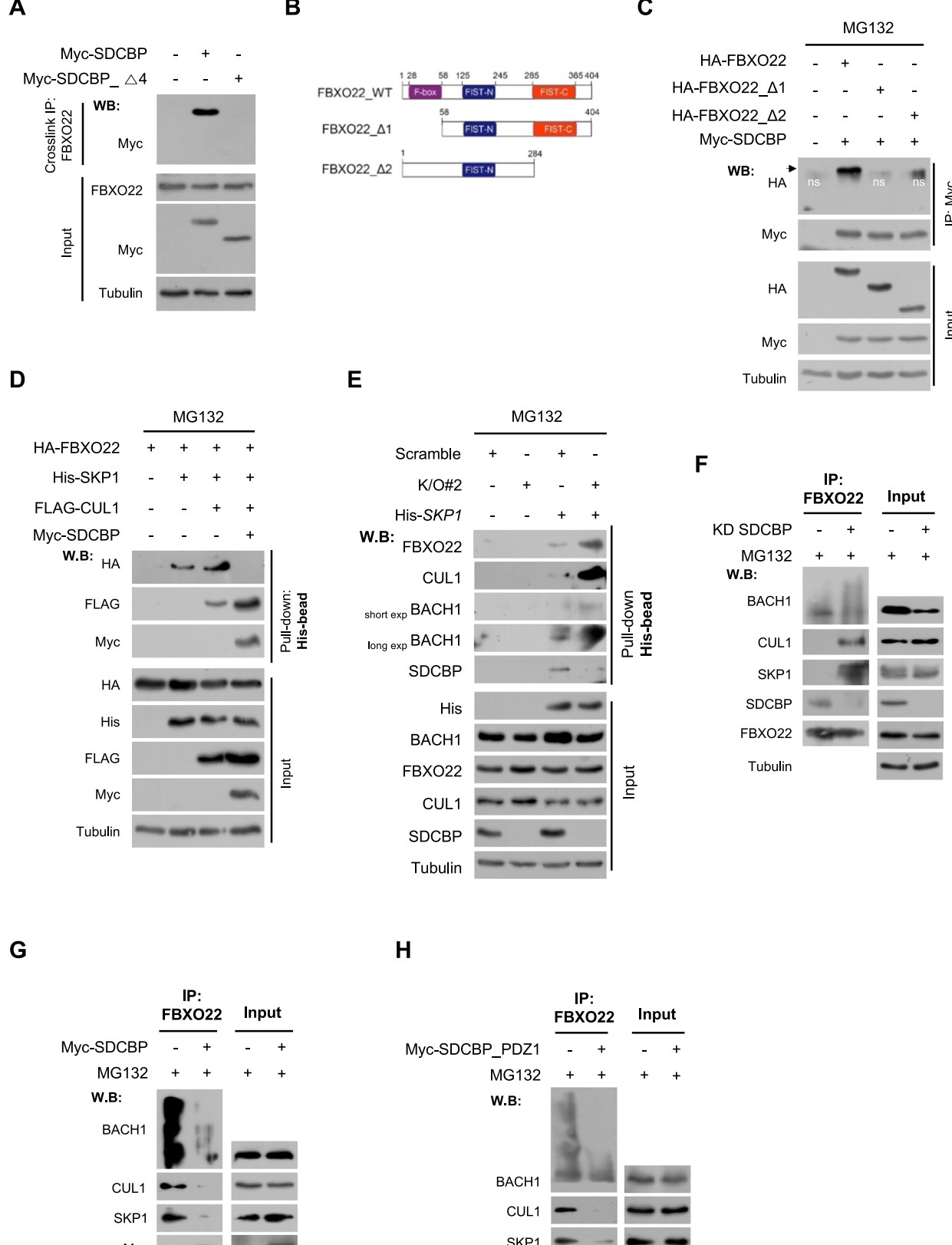

**Figure 4. SDCBP PDZ1 domain plays an essential role in the disruption of SCF$^{FBXO22}$-BACH1 complex.**

(A) Crosslink immunoprecipitation showing an interaction of FBXO22 with SDCBP in Hs578T cells transfected with Myc-SDCBP-expressing vector or Myc-SDCBP_Δ4-expressing vector. (B) Schematic showing the FBXO22 mutant constructs generated using the HA-FBXO22 plasmid. (C) Immunoprecipitation showing SDCBP interactions with FBXO22 and its mutant constructs in HEK293 cells transfected with the indicated plasmids. ns indicates non-specific bands. See also Fig. EV5A,B. (D) His-Pulldown assay showing the effect of SDCBP on SKP1-CUL1-FBXO22 complex formation after the indicated transfections in HEK293 cells. See also Appendix Fig. S1B. (E) His-Pulldown assay showing the effect of SDCBP KO on the SCF$^{FBXO22}$–BACH1 complex formation in the scramble control and SDCBP-KO MDA-MB-231 cells transfected with control vector or His-SKP1-expressing vector. See also Appendix Fig. S1C. (F) Immunoprecipitation showing the effect of SDCBP KD on the SCF$^{FBXO22}$–BACH1 complex formation in MDA-MB-231 cells transfected with scramble or SDCBP siRNA. (G) Immunoprecipitation showing the effect of SDCBP overexpression on the SCF$^{FBXO22}$–BACH1 complex formation in Hs578T cells transfected with control vector or Myc-SDCBP-overexpressing vector. (H) Immunoprecipitation showing the effect of SDCBP PDZ1 domain on the SCF$^{FBXO22}$–BACH1 complex formation in Hs578T cells transfected with a control vector or a Myc-SDCBP-PDZ1-overexpressing vector. Source data are available online for this figure.

SKP1-CUL1-SDCBP complex (Fig. 4D; Appendix Fig. S1B). Consistently, SDCBP KO resulted in a significant increase in SCF$^{FBXO22}$–BACH1 complex formation compared with that in the control cells (Fig. 4E; Appendix Fig. S1C). SDCBP KD promoted the formation of the SCF$^{FBXO22}$–BACH1 complex in MDA-MB-231 cells (Fig. 4F); however, the SDCBP overexpression reversed this effect in Hs578T cells (Fig. 4G). The SDCBP PDZ1 domain also impaired SCF$^{FBXO22}$–BACH1 complex formation (Fig. 4H), further confirming that the PDZ1 domain may play an essential role in SDCBP-mediated BACH1 stabilization by disrupting SCF$^{FBXO22}$–BACH1 complex formation in TNBC cells. These results indicated that the SDCBP adapter stabilizes BACH1 by disrupting the association of the SKP1-CUL1 complex with FBXO22.

FBXO22 targets several substrates, including PD-L1 and PTEN, for degradative ubiquitination in cancer cells (Ge et al, 2020; De et al, 2021). The effects of SDCBP on the expression of other FBXO22 substrates, such as PTEN and PD-L1, were further investigated. SDCBP KO or KD in MDA-MB-231, A549, and NCI-H1299 cells decreased the expression levels of PD-L1 and PTEN (Fig. EV5F–H), whereas the SDCBP overexpression in Hs578T cells reversed these effects (Fig. EV5I). Moreover, SDCBP impaired FBXO22-mediated K48-linked polyubiquitination of PD-L1 (Fig. EV5J), indicating that SDCBP may be a negative regulator of SCF$^{FBXO22}$-substrate complex formation, resulting in the blocking of polyubiquitination and degradation of FBXO22 substrates, including BACH1 and PD-L1.

## SDCBP-BACH1 axis regulates the expression of ETC genes NDUFA4 and COX6B2, and mitochondrial activity in TNBC cells

BACH1 functions as a transcription suppressor of the electron transport chain (ETC) genes, and its depletion has been shown to improve the sensitivity of TNBC cells to ETC inhibitors (Lee et al, 2019). To better understand the regulatory role of SDCBP on BACH1 stabilization, we investigated whether SDCBP regulates the expression levels of BACH1-regulated ETC genes in MDA-MB-231 cells. Exploring target genes regulated by the SDCBP in MDA-MB-231 cells using RNA sequencing identified 1101 upregulated genes in KO#2 cells (FC ≥ 2, $P ≤ 0.05$) as compared with those in the control cells (Appendix Fig. S2A,B). Among these, 382 candidates were BACH1 target genes, including several ETC genes such as NDUFA4, NDUFC2, NDUFA4L2, and COX6B2 (Appendix Fig. S2B). Notably, mRNA expression levels for the three ETC genes NDUFA4, NDUFC2, and COX6B2 were markedly upregulated in SDCBP-KO MDA-MB-231 cells (Fig. 5A). We selected NDUFA4

and COX6B2 for further study. SDCBP re-expression blocked the upregulation of NDUFA4 and COX6B2 expression induced by SDCBP KO (Fig. 5B). Consistently, SDCBP KD increased the protein expression levels for NDUFA4 and COX6B2 in MDA-MB-231 cells, whereas BACH1 overexpression in SDCBP-KD cells reversed this effect (Fig. 5C). Moreover, chromatin immunoprecipitation (ChIP) assays revealed that SDCBP KO reduced the enrichment of BACH1 at the promoters of NDUFA4 and COX6B2 (Fig. 5D; Appendix Fig. S2C). Consistent with the upregulation of ETC genes, SDCBP KD or KO increased the mitochondrial membrane potential and oxygen consumption rate in MDA-MB-231 cells (Fig. 5E,F; and Appendix Fig. S2D). SDCBP re-expression reversed the increase in mitochondrial membrane potential induced by SDCBP KO (Appendix Fig. S2E). These results suggest that the SDCBP-BACH1 axis suppresses mitochondrial respiration by repressing ETC genes, including NDUFA4 and COX6B2, in TNBC cells.

To further investigate the relationship between NDUFA4 or COX6B2 mRNA expression and SDCBP or BACH1 mRNA expression, we analyzed the TCGA dataset. Our results indicated an inverse correlation between NDUFA4 and COX6B2 with SDCBP and BACH1 in TNBC (Appendix Fig. S3A). Kaplan–Meier analyses revealed that high expression of NDUFA4 or COX6B2 mRNA was associated with favorable survival of TNBC patients (Appendix Fig. S3B). We then used a human TNBC tissue microarray to examine the relationship between NDUFA4, SDCBP, and BACH1. Immunohistochemistry staining showed a negative correlation between the expression levels of NDUFA4 and SDCBP, as well as between NDUFA4 and BACH1 (Fig. 5G,H). These results suggest that the SDCBP-BACH1 axis may play a role in tumor progression and outcome in TNBC patients by modulating ETC genes such as NDUFA4 and COX6B2.

## Targeting of SDCBP enhances the anti-cancer effects of ETC inhibitors and increases the anti-tumor activity of metformin in a mouse 4T1 breast cancer model

Further investigations were conducted to assess whether SDCBP KO increases the sensitivity of TNBC cells to the published ETC inhibitors, including metformin, antimycin-A, and oligomycin (Weinberg and Chandel, 2015; Wheaton et al, 2014; Murai and Miyoshi, 2016; Symersky et al, 2012). The results showed that SDCBP KO or KD significantly improved the inhibitory effects of these ETC inhibitors on cell proliferation, cell viability, and colony formation in MDA-MB-231 cells (Fig. 6A–C; Appendix Fig. S4A–D). These results suggested that targeting SDCBP may

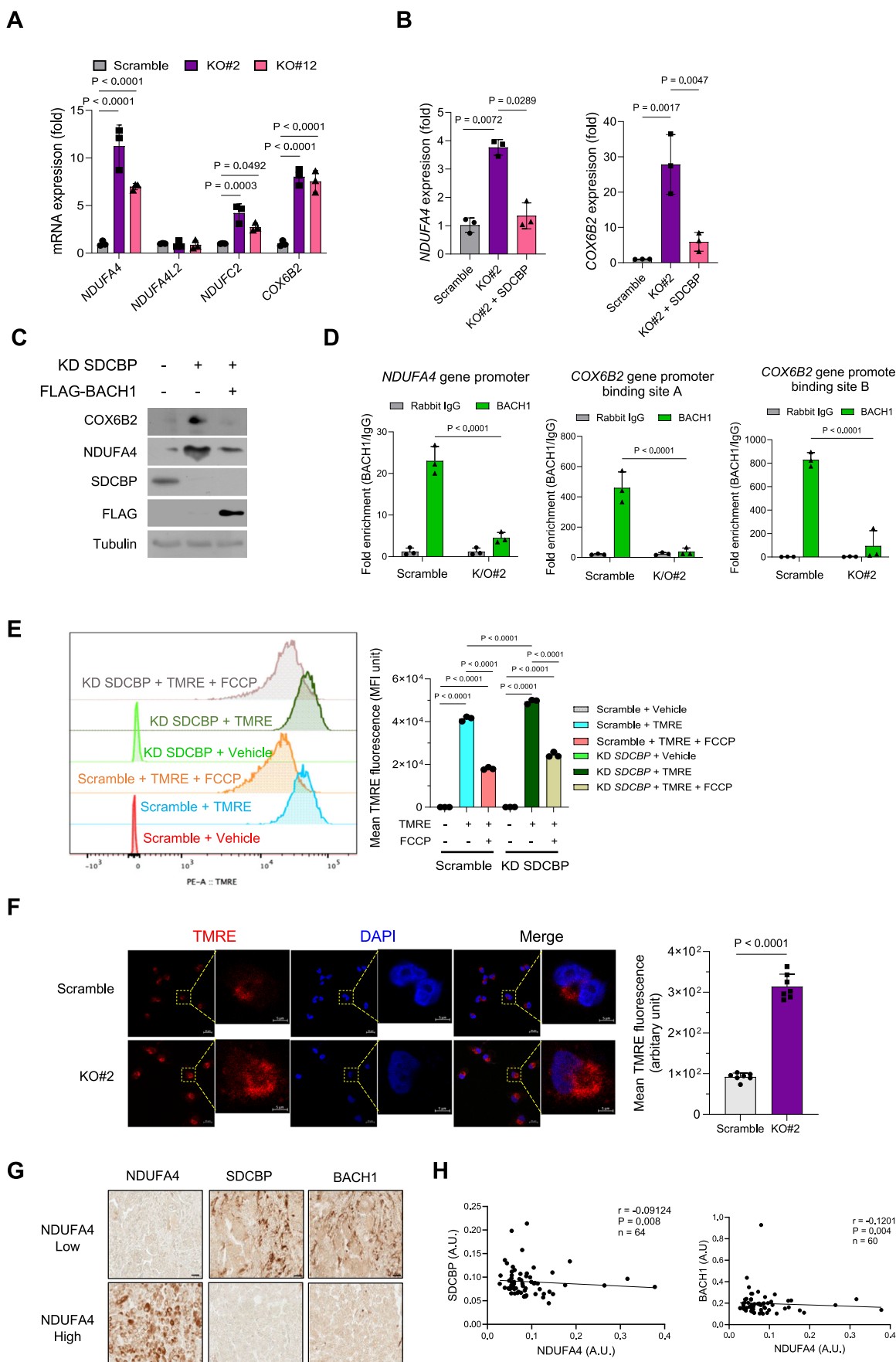

**Figure 5. SDCBP-BACH1 axis regulates the expression of ETC genes NDUFA4 and COX6B2, and mitochondrial activity in TNBC cells.**

(A) Real-time qPCR showing the mRNA expression of BACH1-regulated ETC genes (*NDUFA4, NDUFA4L2, NDUFC2,* and *COX6B2*) in scramble and SDCBP-KO MDA-MB-231 cells (*n* = 3); See also Appendix Fig. S2A,B. (B) Real-time qPCR showing the mRNA expression of *NDUFA4 and COX6B2* in scramble, SDCBP-KO MDA-MB-231 cells, and SDCBP-KO MDA-MB-231 cells transfected with SDCBP (*n* = 3). (C) Western blots showing the expression of mitochondrial proteins NDUFA4 and COX6B2 in MDA-MB-231 cells transfected with SDCBP siRNA in the presence or absence of FLAG-BACH1-expressing vector. (D) ChIP-qPCR showing BACH1 enrichments in the promoter regions of *NDUFA4* and *COX6B2* in the scramble and SDCBP-KO MDA-MB-231 cells. Quantitative data were normalized to IgG binding expression (*n* = 3); See also Appendix Fig. S2C. (E) Left, flow cytometry histogram showing the mitochondrial membrane potentials using TMRE (tetramethylrhodamine ethyl ester) staining in MDA-MB-231 cells transfected with a scramble or SDCBP siRNA after the indicated treatments. FCCP (trifluoromethoxy carbonylcyanide phenylhydrazone). Right, quantification of TMRE fluorescence intensity (*n* = 3). (F) Left, immunofluorescence staining and confocal imaging of the fluorescent signals for TMRE (orange-red color) in the scramble control and SDCBP-KO MDA-MB-231 cells after incubation with TMRE. DAPI (blue color) was used to stain the nucleus (*n* = 7); Representative confocal images are shown; scale bars = 20 μm and 5 μm. Right, fluorescent levels of the TMRE were quantified based on their spectral densities. (G) Representative images of immunohistochemistry staining against the NDUFA4, BACH1, and SDCBP proteins showing a negative correlation between the expression of NDUFA4 and SDCBP in the same sections of TNBC tumor tissues. Scale bar = 20 μm. (H) Pearson correlation coefficients between SDCBP and NDUFA4 protein expression (*n* = 64), and between BACH1 and NDUFA4 protein expression (*n* = 60) in Fig. 5G. Data are expressed as the mean ± SEM and analyzed using two-way ANOVA (**A, D, F**), two-tailed Student's *t* test (**B**), or one-way ANOVA (**E**). *P* values less than 0.05 were considered statistically significant. All experiments were repeated at least three times unless otherwise indicated. Source data are available online for this figure.

improve the sensitivity of TNBC cells to ETC inhibitors, including metformin, by upregulating mitochondrial genes and activity.

Metformin, an FDA-approved medication for type 2 diabetes, has been shown to exert anti-cancer effects against various malignancies (Pernicova and Korbonits, 2014). To further elucidate whether targeting SDCBP enhances the anti-cancer activity of metformin in a TNBC model, murine 4T1 cells were infected with adenoviral SDCBP shRNA, resulting in the complete loss of SDCBP and BACH1 expression (Fig. EV1J). Metformin and/or adenoviral SDCBP shRNA were administered to 4T1 tumor-bearing BALB/c mice when the tumor reached an average volume of 100 mm³. Intra-tumoral injection of the adenoviral SDCBP shRNA effectively suppressed SDCBP expression (Appendix Fig. S4E) as well as 4T1 tumor growth, whereas treatment with metformin alone did not significantly suppress 4T1 tumor growth (Fig. 6D–F; Appendix Fig. S4F). Intriguingly, the combined treatment of metformin and adenoviral SDCBP shRNA significantly enhanced the anti-tumor effects of metformin without decreasing body weight (Fig. 6D–F; Appendix Fig. S4F,G). Immunohistochemical staining revealed that treatment of the 4T1 tumor with adenoviral SDCBP shRNA significantly decreased the expression levels of SDCBP, BACH1, and Ki67, but increased the expression levels of NDUFA4 and COX6B2 compared with those in the control and metformin-treated groups (Fig. 6G). Taken together, these results suggested that targeting SDCBP might be a novel strategy to enhance the anti-tumor efficacy of metformin when used as a therapeutic for TNBC.

## Discussion

Here we investigated that SDCBP abundant, which is associated with a poor prognosis in TNBC patients, leads to BACH1 stabilization by disrupting the SCF^FBXO22–BACH1 complex through its PDZ1 domain, resulting in aggressive TNBC. In detail, SDCBP overexpression caused the accumulation of BACH1 and increased expressions of BACH1-regulated pro-metastatic genes, ultimately driving tumor progression in TNBC cells. In contrast, SDCBP deletion or silencing triggered SCF^FBXO22–BACH1 complex formation, leading to BACH1 degradation via a proteasomal pathway. Additionally, loss of SDCBP downregulated the transcriptions of BACH1-targeted pro-metastatic genes, thereby suppressing tumor progression in TNBC cells. Furthermore, loss

of SDCBP significantly upregulated the transcriptions of BACH1-repressed ETC genes, including *NDUFA4* and *COX6B2*, and induced mitochondrial respiration in TNBC cells (Fig. 7).

The SCF complex is assembled by recruitment of SKP1 and Rbx1/CUL1, to different F-box proteins following substrate selectivity (Wang et al, 2014). Our results showed FBXO22 KD significantly induced BACH1 expression and prolonged the BACH1 half-life, indicating that FBXO22 is the critical E3 ubiquitin ligase targeting BACH1 into proteasomal degradation in TNBC cells. Although FBXO22 was proposed as a specific ubiquitin E3 for SCF-mediated BACH1 degradation in lung cancer cells, experimental evidence for FBXO22-mediated BACH1 ubiquitination in vitro have not yet been obtained (Lignitto et al, 2019). In this study, we successfully identified the K48-linked polyubiquitin signal of FBXO22-mediated BACH1 degradative ubiquitination in vitro. Moreover, we identified that SDCBP may disassociate the SCF^FBXO22–BACH1 complex, resulting in an abundance of BACH1 in TNBC cells. Indeed, SDCBP KO or KD induced BACH1 K48-linked polyubiquitination and proteasomal degradation, whereas SDCBP overexpression completely reversed these effects. Importantly, in vitro ubiquitination assays confirmed that FBXO22 ubiquitylated BACH1, and SDCBP blocked FBXO22-mediated BACH1 degradative ubiquitination. Mechanistically, SDCBP interacts with FBXO22, then impairs the association of FBXO22 to SKP1-CUL1 via its PDZ1 domain, resulting in the dissociation of the SCF^FBXO22–BACH1 complex. FBXO22 also regulates the expression of several tumorigenic proteins including PD-L1, via SCF^FBXO22-targeted substrate for proteasomal degradation in cancer cells (De et al, 2021). In this study, SDCBP was also found to inhibit FBXO22-mediated degradative polyubiquitination of PD-L1. The loss of SDCBP decreased PD-L1 expression in TNBC cells; however, the overexpression of SDCBP increased PD-L1 expression. SDCBP may protect SCF^FBXO22-targeted substrates, including BACH1 and PD-L1, from proteasome degradation in TNBC cells. Consistent with our results, SDCBP has been reported to induce PD-L1 abundance in TNBC cells (Liu et al, 2018). To the best of our knowledge, the role of SDCBP and its PDZ1 domain as regulators of the SCF-substrate complex formation has not been investigated previously. The current study is the first to report that the PDZ domain protein regulates SCF^FBXO22-substrate complex formation. However, the protein structure-based mechanisms by which SDCBP regulates SCF^FBXO22–BACH1 complex formation require further elucidation.

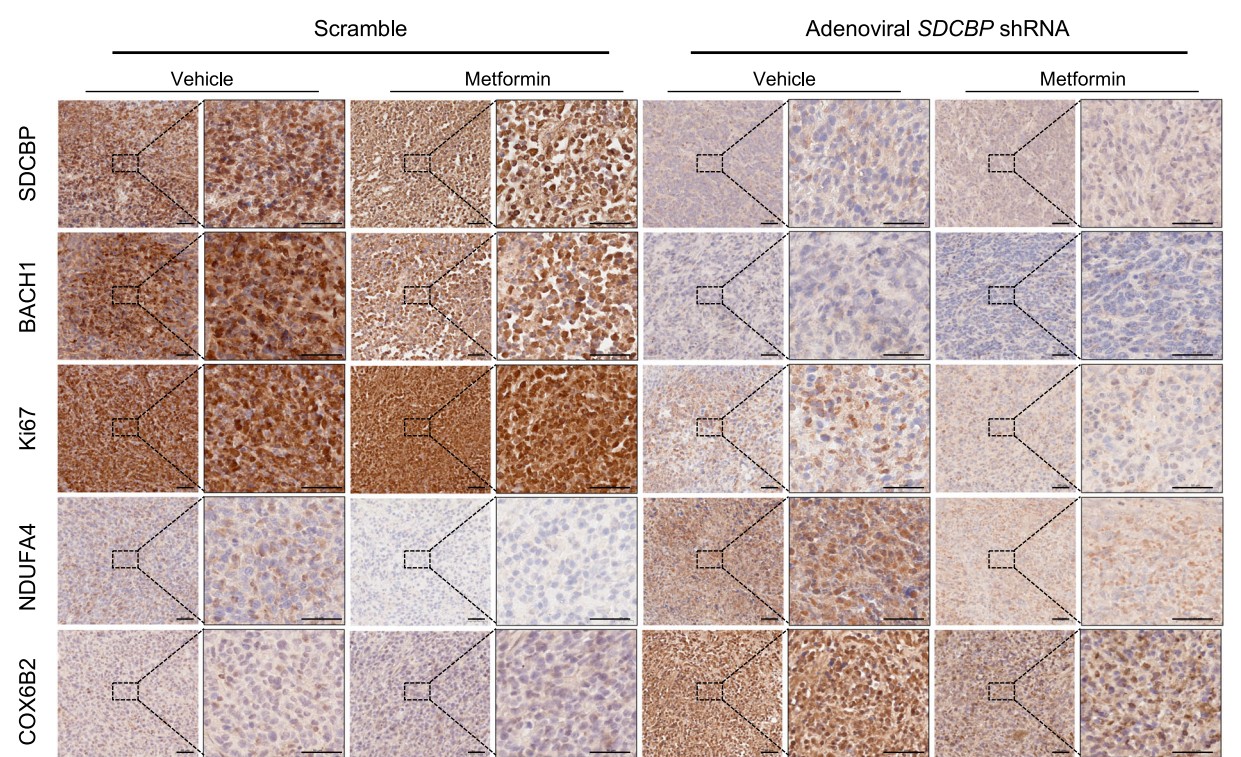

◄ **Figure 6. Targeting of SDCBP enhances the anti-cancer effects of ETC inhibitors and increases the anti-tumor activity of metformin in a mouse 4T1 breast cancer model.**

(A) Cell proliferation of MDA-MB-231 cells transfected with a scramble or SDCBP siRNA after treatment with metformin for the indicated periods of time ($n = 3$). (B) Cell viability of MDA-MB-231 cells transfected with a scramble or SDCBP siRNA after treatment with the indicated concentrations of metformin for 96 h ($n = 5$). (C) Colony formation for MDA-MB-231 cells transfected with a scramble or SDCBP siRNA after treatment with the indicated concentrations of metformin. Colony numbers were counted and converted to percentages by normalizing with the control groups ($n = 3$). (D–F) Effect of the indicated treatment on 4T1 tumor growth. Tumor growth was monitored in BALB/c mice bearing 4T1 cells after mammary fat-pad injections. When the average tumor volumes reached 100 mm³, the mice ($n = 7$ mice/group) were administered with 100 mg/kg metformin (once a day) and/or adenoviral SDCBP shRNA ($1 \times 10^9$ PFU/mice). Black arrows indicate the day of adenoviral SDCBP shRNA injection. Final tumor volume (E) and weight (F) are shown. (G) Immunohistochemistry staining against SDCBP, BACH1, Ki67, NDUFA4, and COX6B2 protein in 4T1 tumors from BALB/c mice in Fig. 6A. Representative images of the IHC staining are shown. Scale bar = 50 μm for low (left) and high (right) magnification. Data are expressed as the mean ± SEM and analyzed using two-way ANOVA (A–C), one-way ANOVA (E), or two-tailed Student's $t$ test (F). $P$ values less than 0.05 were considered statistically significant. All experiments were repeated at least three times unless otherwise indicated. Source data are available online for this figure.

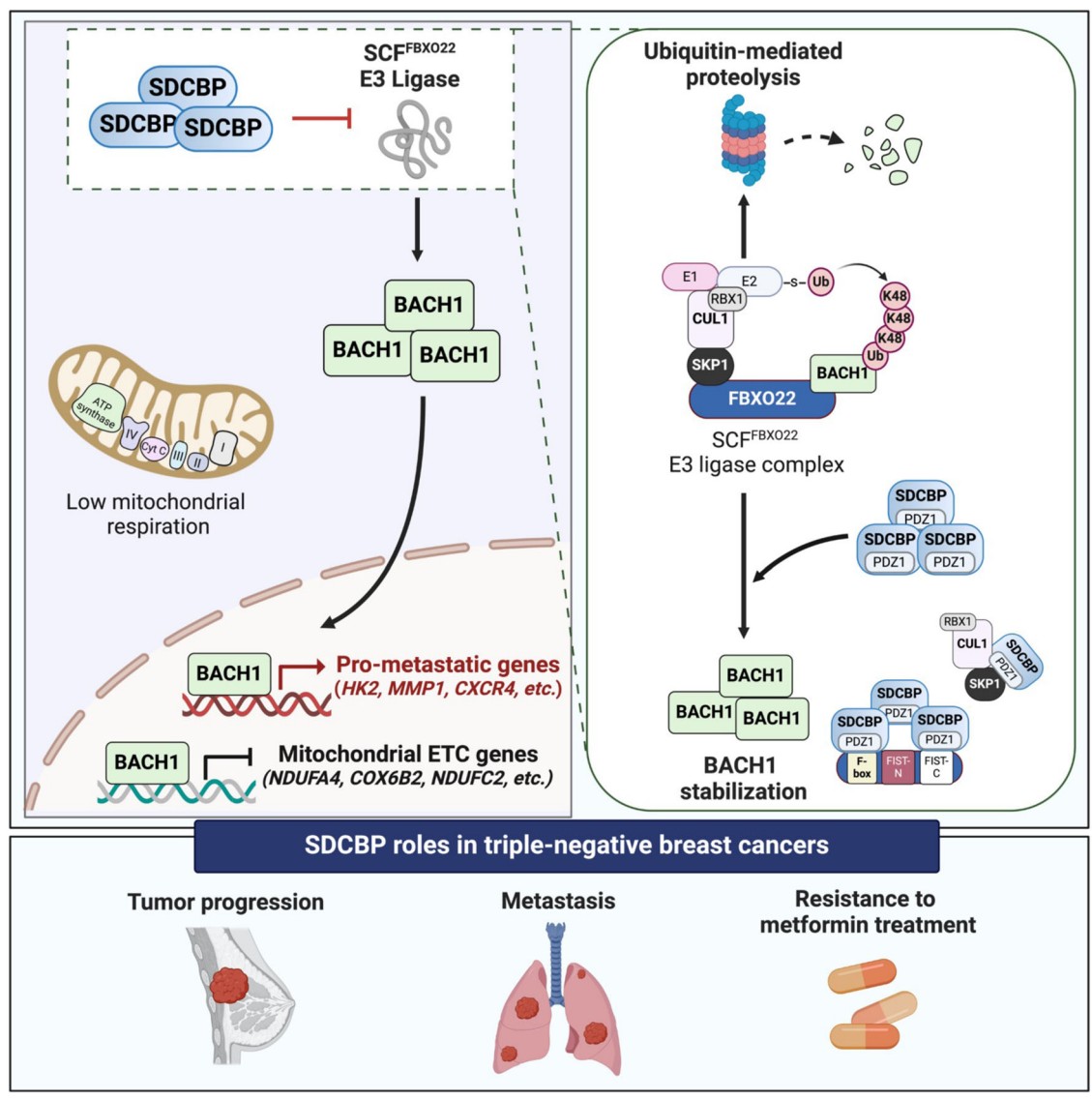

**Figure 7. Schematic summary of this study.**

Schematic showing the novel oncogenic roles of SDCBP in promoting aggressiveness and mitochondrial inhibitor resistance in TNBCs. An abundance of SDCBP stabilizes the BACH1 protein by blocking E3 ubiquitin ligase SCF$^{FBXO22}$ complex-targeted BACH1 for degradative ubiquitination. Mechanistically, SDCBP binds to different members of the SCF$^{FBXO22}$ complex, including SKP1 and FBXO22, via its PDZ1 domain and induces SCF$^{FBXO22}$ complex disassociation, suggesting that SDCBP is a key adapter regulating the activity of the E3 ubiquitin ligase SCF$^{FBXO22}$ complex in the proteasomal pathway. SDCBP-induced BACH1 accumulation upregulates several pro-metastatic genes and downregulates numerous mitochondrial ETC genes, resulting in tumor progression and high resistance to metformin treatment in TNBCs. Targeting SDCBP switches the SCF$^{FBXO22}$ complex to degrade BACH1 protein via the proteasome, reducing tumor aggressiveness and boosting the anti-tumor effect of metformin administration in TNBCs. Source data are available online for this figure.

BACH1 stability is sensitive to heme or hemin, which are essential for E3 ubiquitin ligases, including HOIL-1 and FBXO22, mediated BACH1 degradation via proteasome in lung cancer cells (Lignitto et al, 2019; Suzuki et al, 2004; Zenke-Kawasaki et al, 2007). In previous studies, NRF2 activation stabilized BACH1 protein via inducing HO-1 for heme degradation, leading to high levels of metastasis in lung cancers (Lignitto et al, 2019). Interestingly, our results showed that HO-1 or NRF2 did not significantly change BACH1 expression, indicating that BACH1 stability is independent of the NRF2/HO-1 pathway in TNBC cells. Our in vitro ubiquitination data showed that SDCBP completely blocks the degradative ubiquitination of the SCF$^{FBXO22}$–BACH1 complex. Notably, SDCBP did not alter the free heme levels in TNBC cells. These results suggested that SDCBP-mediated BACH1 stabilization may be an alternative BACH1 stabilization mechanism, regardless of the NRF-2/HO-1 levels in TNBC cells.

BACH1 suppresses mitochondrial activity by reprogramming the glucose utilization into glycolytic metabolism via upregulating the transcription of glycolytic genes, including *HK2* and *GAPDH*, in lung cancer cells (Wiel et al, 2019; Lee et al, 2019). BACH1, at the same time, also represses mitochondrial respiration by down-regulating the transcription of ETC genes, including *ATP5D*, *COX15*, and *UQCRC1*, in TNBC cells (Lee et al, 2019). Consistently, our data revealed that SDCBP KO in MDA-MB-231 cells significantly increased the mRNA levels of three ETC genes, namely *NDUFC2, COX6B2*, and *NDUFA4*. Overexpression of BACH1 in SDCBP-KD cells abolished the SDCBP KO-induced expression of *NDUFC2, COX6B2*, and *NDUFA4*. Moreover, SDCBP KO resulted in the significant suppression of BACH1 binding to the promoter regions of these ETC genes. Consistent with the upregulation of these ETC genes in the SDCBP-KO TNBC cells, SDCBP KD also increased mitochondrial oxygen consumption rate and mitochondrial membrane potential. Intriguingly, targeting SDCBP enhanced the sensitivity of MDA-MB-231 cells to mitochondrial ETC inhibitors including metformin, antimycin-A, and oligomycin. These results confirmed that SDCBP is a positive regulator of BACH1 expression, and it was further proposed that SDCBP abundant may suppress mitochondrial respiration in TNBC cells. The major hindrance in TNBC treatment is the lack of effective therapeutic options owing to the limited number of targets (Yin et al, 2020; Copel and Kanaan, 2021; Bianchini et al, 2016). Thus, this study shows that targeting SDCBP, particularly its PDZ1 domain, could be a novel alternative approach to block the BACH1 pathway and optimize anti-tumor effectiveness of FDA-approved mitochondrial inhibitor metformin as a new combinational strategy for treating patients with aggressive TNBC (Fig. 7).

# Methods

**Reagents and tools table**

| REAGENT or RESOURCE | SOURCE | IDENTIFIER |
|---|---|---|
| **Antibodies** | | |
| BACH1 (F-9), Mouse monoclonal antibody (WB, 1/500; IHC, 1/500) | Santa Cruz Biotechnology | Cat#sc-271211 |
| β-actin, Rabbit antibody (WB, 1/1000) | Cell Signaling Technology | Cat#4967 |
| COX6B2, Mouse monoclonal antibody (WB, 1/500) | OriGene | Cat#TA501377 |
| CUL-1 (D-5), Mouse monoclonal antibody (WB, 1/1000) | Santa Cruz Biotechnology | Cat#sc-17775 |
| FBXO22 (FF-7), Mouse monoclonal antibody (WB, 1/1000) | Santa Cruz Biotechnology | Cat#sc-100736 |
| FLAG (M2), Mouse monoclonal antibody (WB, 1/500) | Sigma-Aldrich | Cat#F3165 |
| HA-Tag (C29F4), Rabbit monoclonal antibody (WB, 1/1000) | Cell Signaling Technology | Cat#3724 |
| HA-Tag (12CA5), Mouse monoclonal antibody (WB, 1/500) | Addgene | Cat#194541 |
| His-Tag (H-3), Mouse monoclonal antibody (WB, 1/1000) | Santa Cruz Biotechnology | Cat#sc-8036 |
| His-Tag, Rabbit antibody (WB, 1/1000) | Cell Signaling Technology | Cat#2365 |
| HO-1/HMOX1 (D-8) Mouse monoclonal antibody (WB, 1/1000) | Santa Cruz Biotechnology | Cat#sc-136961 |
| HOIL-1/RBCK1 (E-2) Mouse monoclonal antibody (WB, 1/1000) | Santa Cruz Biotechnology | Cat#sc-365523 |
| c-Myc (9E10), Mouse monoclonal antibody (WB, 1/2000) | Santa Cruz Biotechnology | Cat#sc-40 |
| NDUFA4, Rabbit polyclonal antibody (WB, 1/500; IHC, 1/500) | Invitrogen | Cat#PA5-51021 |
| NRF2 (EP1808Y), Recombinant Rabbit monoclonal antibody (WB, 1/500) | Abcam | Cat#ab62352 |
| PD-L1, Goat polyclonal antibody (WB, 1/500) | Invitrogen | Cat#PA5-18337 |
| PTEN (138G6) Rabbit monoclonal antibody (WB, 1/500) | Cell Signaling Technology | Cat#9559 |
| Skp1/p19 (H-163), Mouse monoclonal antibody (WB, 1/1000) | Santa Cruz Biotechnology | Cat#sc-7163 |
| Syntenin-1 (2C12), Mouse monoclonal antibody (WB, 1/5000) | Abnova | Cat#H00006386-M01 |
| HRP-conjugated-α-Tubulin, Mouse monoclonal antibody (WB, 1/10,000) | Protein-tech | Cat#HRP-66031 |
| Ubiquitin (P4D1), Mouse monoclonal antibody (WB, 1/250) | Santa Cruz Biotechnology | Cat#sc-8017 |
| HRP-linked anti-rabbit IgG (WB, 1/1000) | Cell Signaling Technology | Cat#7074 |

| REAGENT or RESOURCE | SOURCE | IDENTIFIER |
|---|---|---|
| HRP-linked anti-mouse IgG (WB, 1/1000) | Cell Signaling Technology | Cat#7076 |
| Mouse HRP-linked anti-goat IgG (WB, 1/2500) | Santa Cruz Biotechnology | Cat#sc-2354 |
| IgG, Rabbit polyclonal antibody (IP) | Protein-tech | Cat#30000-0-AP |
| FBXO22 (FF-7), Mouse monoclonal antibody (IP) | Santa Cruz Biotechnology | Cat#sc-100736 |
| FLAG (M2), Mouse monoclonal antibody (IP) | Sigma-Aldrich | Cat#F3165 |
| HA-Tag (C29F4), Rabbit monoclonal antibody (IP) | Cell Signaling Technology | Cat#3724 |
| His-Tag (H-3), Mouse monoclonal antibody (IP/Pulldown) | Santa Cruz Biotechnology | Cat#sc-8036 |
| His-Tag, Rabbit antibody (IP) | Cell Signaling Technology | Cat#2365 |
| c-Myc (9E10), Mouse monoclonal antibody (IP) | Santa Cruz Biotechnology | Cat#sc-40 |
| Skp1/p19 (H-163), Mouse monoclonal antibody (IP) | Santa Cruz Biotechnology | Cat#sc-7163 |
| Human BACH1, Goat polyclonal antibody (CLSM, 1/500) | R&D Systems® | Cat#AF5776 |
| Human BACH1, Rabbit monoclonal antibody (CLSM, 1/500) | Cell Signaling Technology | Cat#4578 |
| Syntenin-1, Rabbit polyclonal antibody (CLSM, 1/500) | Protein-tech | Cat#22399-1-AP |
| Syntenin-1 (2C12), Mouse monoclonal antibody (CLSM, 1/10 000; IHC, 1/500) | Abnova | Cat#H00006386-M01 |
| FBXO22 (FF-7), Mouse monoclonal antibody (CLSM, 1/500) | Santa Cruz Biotechnology | Cat#sc-100736 |
| Goat anti-Rabbit IgG (H + L) Alexa Fluor® 488 (CLSM, 1/1000) | Invitrogen | Cat#A-11008 |
| Goat anti-Rabbit IgG (H + L) Alexa Fluor® 546 (CLSM, 1/1000) | Invitrogen | Cat#A-11071 |
| Donley anti-Goat IgG (H + L) Alexa Fluor® 546 (CLSM, 1/1000) | Invitrogen | Cat#A-11056 |
| Donkey anti-Goat IgG (H + L) Alexa Fluor® 647 (CLSM, 1/1000) | Abcam | Cat#ab150131 |
| Goat anti-Mouse IgG (H + L) Alexa Fluor® 488 (CLSM, 1/1000) | Invitrogen | Cat#A-11001 |
| Goat anti-Mouse IgG (H + L) Alexa Fluor® 647 (CLSM, 1/1000) | Invitrogen | Cat#A-21235 |
| PD-L1 (EPR19759), Recombinant Rabbit monoclonal antibody (IHC) | Abcam | Cat#ab213524 |
| Ki-67 (SP6), Recombinant Rabbit monoclonal antibody (IHC, 1/500) | Invitrogen | Cat#MA5-14520 |
| humanBACH1, Goat polyclonal antibody (TMA, 1/500) | R&D Systems® | Cat#AF5776 |
| Syntnein-1, Rabbit monoclonal antibody (TMA, 1/500) | ABclonal | Cat#A5497 |
| BACH1 (F-9), Mouse monoclonal antibody (ChIP-qPCR) | Santa Cruz Biotechnology | Cat#sc-271211 |
| Anti-RNA Polymerase II antibody (ChIP-qPCR) | Thermo Scientific™ | Cat#1862243 |
| Normal Rabbit IgG antibody | Thermo Scientific™ | Cat#1862244 |
| **Bacteria and virus strains** | | |
| Ad-LacZ-adenovirus | GENENMED | Cat#ADS19-015 |
| Ad-m-SDCBP-shRNA-silencing-adenovirus | VECTOR BIOLABS | Cat#shADV271-494 |
| HIT-Competent Cells-DH5α | RBC Bioscience | Cat#RH617 |
| One Shot™ Stbl3™ Chemically Competent E. coli | Invitrogen | Cat#C737303 |
| **Reagents and recombinant proteins** | | |
| Fetal Bovine Serum | Biowest | Cat#S1480 |
| Penicillin & Streptomycin | Gibco | Cat#15140-122 |
| Dulbecco's modified Eagle medium (DMEM) | Biowest | Cat#L0103-500 |
| Roswell Park Memorial Institute 1640 (RPMI-1640) | Biowest | Cat#L0498-500 |
| Phosphate Buffered Saline (PBS) | Gibco | Cat#10010-023 |
| 0.05% Trypsin-EDTA | Gibco | Cat#25300-062 |
| Puromycin dihydrochloride | Sigma-Aldrich | Cat#P8833 |
| MG132 | Sigma-Aldrich | Cat#C2211 |
| Cycloheximide (CHX) | Sigma-Aldrich | Cat#C7698 |
| Hemin | Sigma-Aldrich | Cat#H9039 |
| Metformin Hydrochloride | Sigma-Aldrich | Cat#PHR1084 |

| REAGENT or RESOURCE | SOURCE | IDENTIFIER |
|---|---|---|
| RIPA buffer | Sigma-Aldrich | Cat#R0278 |
| cOmplete™ protease inhibitor cocktail | Roche-Diagnostics GmbH | Cat#11697498001 |
| Bis³ (bis(sulfosuccinimidyl) suberate) | Thermo Scientific™ | Cat#21580 |
| MTT (3-(4, 5-dimethylthiazol-2-yl)-2,5-diphenyltetrazolium bromide) | Duchefa Biochemie | Cat#M1415 |
| Trypan Blue Stain | Invitrogen | Cat#T10282 |
| Methyl alcohol | Sigma-Aldrich | Cat#34860 |
| Paraformaldehyde | Sigma-Aldrich | Cat#P6148 |
| Crystal Violet | Sigma-Aldrich | Cat#C0775 |
| Hematoxylin Solution | Abcam | Cat#ab220365 |
| Eosin Y disodium slat | Sigma-Aldrich | Cat#E4382 |
| TMRE (tetramethylrhodamine, ethyl ester) | Invitrogen | Cat#T669 |
| FCCP (carbonyl cyanide 4-(trifluoromethoxy)phenylhydrazone) | Thermo Scientific™ | Cat#04-531-0 |
| Triton™ X-100 | Sigma-Aldrich | Cat#9036-165 |
| DAPI (2-(4-amidinophenyl)-6-indolecarbamidine) | Sigma-Aldrich | Cat#D9542 |
| Dimethyl Sulfoxide | Sigma-Aldrich | Cat#D2660 |
| Hemin | Sigma-Aldrich | Cat#H9039 |
| Dithiothreitol | Duchefa Biochemie | Cat#D1309 |
| Fibronectin | Sigma-Aldrich | Cat#F1141 |
| Chloroquine diphosphate salt | Sigma-Aldrich | Cat#C6628 |
| Hygromycin B | Sigma-Aldrich | Cat#H0654 |
| Ampicillin Sodium | Duchefa Biochemie | Cat#A0104 |
| Kanamycine sulfate monohydrate | Duchefa Biochemie | Cat#K0126 |
| Biological Human Syntenin-1 protein | Sino Biological | Cat#15640-H07E |
| Human BACH1 Recombinant protein | Origene | Cat#TP321628 |
| Human UbcH5A Recombinant protein | Abcam | Cat#ab269096 |
| **Critical commercial assays** | | |
| Lipofectamine™ 3000 Transfection Reagent | Invitrogen | Cat#L3000015 |
| RNeasy® Mini Kit | QIAGEN | Cat#74106 |
| Maxime RT-PreMix kit | LiliF Diagnostics | Cat#25081 |
| 2X Fast-Q-PCR Master Mix Kit (SYBR, ROX) | SMOBIO | Cat#TQ1210 |
| MicroAmp® Fast Reaction Tubes (8 Tubes/Strip) | Thermo Scientific™ | Cat#4358293 |
| MicroAmp® Optical 8-Cap Strip | Thermo Scientific™ | Cat#4323032 |
| Pierce™ Agarose ChIP Kit | Thermo Scientific™ | Cat#26156 |
| Pierce™ Chromatin Prep Module Kit | Thermo Scientific™ | Cat#26158 |
| Bradford reagent for protein quantification | Sigma-Aldrich | Cat#B6916 |
| EZ-Western Solution chemiluminescence kit | DogenBIO Ltd | Cat#DLS2305 |
| A/G PLUS-agarose bead | Santa Cruz Biotechnology | Cat#sc2003 |
| Dynabeads™ Protein G | Invitrogen | Cat#10003D |
| VeriBlot-Rat mAb to Mouse IgG-HRP for IP | Abcam | Cat#ab131368 |
| VeriBlot for IP Detection Reagent-HRP | Abcam | Cat#ab131366 |
| Dynabeads™ His-Tag Isolation & Pulldown | Invitrogen | Cat#10103D |
| Ubiquitylation Assay kit | Abcam | Cat#ab139467 |
| E2-Ubiquitin Conjugation Kit | Abcam | Cat#ab139472 |
| Heme assay kit | Abcam | Cat#ab272534 |
| Seahorse XFe35 V7 PS kit | Agilent Technologies | Cat#100777-004 |

| REAGENT or RESOURCE | SOURCE | IDENTIFIER |
|---|---|---|
| Dako REAL Envision Detection System Peroxidase/DAB+, Rabbit/Mouse | Agilent DAKO | Cat#K5007 |
| Cell and Tissue Staining Kit Anti-Rabbit HRP-DAB system | R&D Systems® | Cat#CTS005 |
| Cell and Tissue Staining Kit Anti-Goat HRP-DAB system | R&D Systems® | Cat#CTS008 |
| Dako REAL™ Peroxidase-Blocking Solution | Agilent DAKO | Cat#S20286-2 |
| Dako Protein Block Serum-Free Ready-To-Use | Agilent DAKO | Cat#X0909 |
| Triple-negative breast carcinoma and normal tissue microarray | TissueArray-US Biomax | Cat#BR1301; Cat#BR1902 |
| Muta-direct™ site-directed Mutagenesis kit | LiliF Diagnostics | Cat#15071 |
| Hybrid-Q™ Plasmid Rapidprep-mini kit | GeneAll® | Cat#100-102 |
| NucleoBond Xtra-Midi kit for transfection-grade plasmid DNA | MACHEREY-NAGEL | Cat#740410 |
| Expin™ PCR SVmini Kit | GeneAll® | Cat#103-150 |
| Expin™ Gel SV Kit | GeneAll® | Cat#102-102 |
| **Experimental models: cell lines** | | |
| MDA-MB-231 | ATCC | Cat#HTB-26™ |
| 4T1 | ATCC | Cat#CRL-2539™ |
| HEK293 | ATCC | Cat#CRL-1573™ |
| HEK293T | ATCC | Cat#CRL-3216™ |
| MDA-MB-468 | ATCC | Cat#HTB-132™ |
| Hs578T | ATCC | Cat#HTB-126™ |
| MCF-7 | ATCC | Cat#HTB-22™ |
| T-47D | ATCC | Cat#HTB-133™ |
| *CRISPR-Cas9_SDCBP Knockout* MDA-MB-231 | This study | N/A |
| A549 | ATCC | Cat#CCL-185™ |
| NCI-H1299 | ATCC | Cat#CRL-5803™ |
| **Deposited data** | | |
| RNA sequencing data *SDCBP Knockout/SDCBP scramble* | This study | Macrogen NGS Analysis number: HN00172576 |
| **Experimental models: organisms/strains** | | |
| NCr-BALB/c Nude mice (Female) | DBL Korea | N/A |
| BALB/c mice (Female) | DBL Korea | N/A |
| **Oligonucleotides** | | |
| *List primers for CRISPR-Cas9 deletion assay* | | |
| *Human Knockout SDCBP.gRNA.Puro.v2gN19* (PAGE purify) | Sense: 5′-CACCGCCTTCAAGTCTTCGAGAGA-3′ Anti-sense: 5′-AAACTCTCTCGAAGACTTGAAGGC-3′ | |
| *List primers for ChIP-qPCR assay* | | |
| *NDUFA4 promoter* (PAGE purify) | Forward primer 5′-TGACCTTAGCGGCAAGAGGC-3′ Reverse primer 5′-CTGAGAGCCGGGTCCGC-3′ | |
| *COX6B2 promoter 1st position* (PAGE purify) | Forward primer 5′-ACCACCTGCCCAGATTCCAG-3′ Reverse primer 5′-ATGCCTGGCTTAGGAGAGCG-3′ | |
| *COX6B2 promoter 2nd position* (PAGE purify | Forward primer 5′-GGGACGCTCTCCTAAGCCAG-3′ Reverse primer 5′-TGGGCTCCTCTGGACAGTTT-3′ | |

| REAGENT or RESOURCE | SOURCE | IDENTIFIER |
|---|---|---|
| *List of RT-qPCR primers for human cells* | | |
| *Syntenin-1*<br>(BioRP purify) | Forward primer<br>5′-GCACCGCTCTCTTACACTCG-3′<br>Reverse primer<br>5′-GCCTGAATTACTTTGTCTACCTTCA-3′ | |
| *BACH1*<br>(BioRP purify) | Forward primer<br>5′-CACCGAAGGAGACAGTGAATCC-3′<br>Reverse primer<br>5′-GCTGTTCTGGAGTAAGCTTGTGC-3′ | |
| *GAPDH*<br>(BioRP purify) | Forward primer<br>5′-TGGACTCCACGACGTACTCA-3′<br>Reverse primer<br>5′-TCAAGGCTGAGAACGGGAAG-3′ | |
| *β-actin*<br>(BioRP purify) | Forward primer<br>5′-ACGTTGCTATCCAGGCTGTG-3′<br>Reverse primer<br>5′-GAGGGCATACCCCTCGTAGA-3′ | |
| *HK2*<br>(BioRP purify) | Forward primer<br>5′-ACGGTCTTATGTAGACGCTTGGCA-3′<br>Reverse primer<br>5′-CTGCAGCGCATCAAGGAGAACAAA-3′ | |
| *MMP1*<br>(BioRP purify) | Forward primer<br>5′-CTCTGGAGTAATGTCACACCTCT-3′<br>Reverse primer<br>5′-TGTTGGTCCACCTTTCATCTTC-3′ | |
| *CXCR4*<br>(BioRP purify) | Forward primer<br>5′-CTCCAAGCTGTCACACTCCA-3′<br>Reverse primer<br>5′-TCGATGCTGATCCCAATGTA-3′ | |
| *VEGF*<br>(BioRP purify) | Forward primer<br>5′-TTGCCCTTGCTGCTCTACCTC-3′<br>Reverse primer<br>5′-AGCTGCGCTGATAGACATCC-3′ | |
| *HO-1*<br>(BioRP purify) | Forward primer<br>5′-AGCCTTGCGGTGCAGCTCTTC-3′<br>Reverse primer<br>5′-CCCAGGCAGAGAATGCTGAGTTC-3′ | |
| *NQO1*<br>(BioRP purify) | Forward primer<br>5′-AGTGGTGATGGAAAGCACTGCCTTC-3′<br>Reverse primer<br>5′-GGAGAGTTTGCTTACACTTACGC-3′ | |
| *KEAP1*<br>(BioRP purify) | Forward primer<br>5′-GTGTAGGCGAATTCAATGAGG-3′<br>Reverse primer<br>5′-GTCTTCAAGGCCATGTTCAC-3′ | |
| *MMP13*<br>(BioRP purify) | Forward primer<br>5′-TAGAGAGACCTGGATCCCTTG-3′<br>Reverse primer<br>5′-GTCCGATGTAACTCCTCTGA-3′ | |
| *PDK1*<br>(BioRP purify) | Forward primer<br>5′-ACCTTCTCCCCCATCCTGAG-3′<br>Reverse primer<br>5′-CTGATGCCCAAGCCCACCT-3′ | |
| *PDHA*<br>(BioRP purify) | Forward primer<br>5′-AACACAATGGGTTCCTGAGC-3′<br>Reverse primer<br>5′-CGAGAGGACAATGGGAAAAA-3′ | |
| *PDHA1*<br>(BioRP purify) | Forward primer<br>5′-GGTGCAGTTTGATCACCGGAT-3′<br>Reverse primer<br>5′-TGCCCAAGCCCACCTGTGTA-3′ | |

| REAGENT or RESOURCE | SOURCE | IDENTIFIER |
|---|---|---|
| PDHX<br>(BioRP purify) | Forward primer<br>5'-TTGGGAGGTTCCGACCTGT-3'<br>Reverse primer<br>5'-CAACCACTCGACTGTCACTTG-3' | |
| AceCS1<br>(BioRP purify) | Forward primer<br>5'-GGGGCACAAGTGCTAATTGC-3'<br>Reverse primer<br>5'-GTTGTCCAGCAGGCTAACC-3' | |
| ATP5D<br>(BioRP purify) | Forward primer<br>5'-TCCCACGCAGGTGTTCTTC-3'<br>Reverse primer<br>5'-GGAACCGCTGCTCACAAAGT-3' | |
| ATP6V1C2<br>(BioRP purify) | Forward primer<br>5'-AGAAAAACGGGCCAAGGACA-3'<br>Reverse primer<br>5'-TCCTCCTTGGCTTGCTTCAA-3' | |
| ATP6V1G3<br>(BioRP purify) | Forward primer<br>5'-ATCCACCAGCTTCTTCAGGC-3'<br>Reverse primer<br>5'-CCATTGCTTCCTCCTTGGCT-3' | |
| COX15<br>(BioRP purify) | Forward primer<br>5'-CAGCGCCTAGAGCACAGTG-3'<br>Reverse primer<br>5'-GCCAGACTCTGTCAACCTAGT-3' | |
| UQCRC1<br>(BioRP purify) | Forward primer<br>5'-GGGGCACAAGTGCTATTGC-3'<br>Reverse primer<br>5'-GTTGTCCAGCAGGCTAACC-3' | |
| SLC25A15<br>(BioRP purify) | Forward primer<br>5'-CCTGAAGACTTACTCCCAGGT-3'<br>Reverse primer<br>5'-GCGATGTTGGCGATTAGTGC-3' | |
| NDUFA4<br>(BioRP purify) | Forward primer<br>5'-GGAGGAGGGAGAGGACAGAG-3'<br>Reverse primer<br>5'-GAGTCGCAGCAAGTAAAGCG-3' | |
| NDUFA4L2<br>(BioRP purify) | Forward primer<br>5'-GGAGGAGGGAGAGGACAGAG-3'<br>Reverse primer<br>5'-GAGTCGCAGCAAGTAAAGCG-3' | |
| NDUFC2<br>(BioRP purify) | Forward primer<br>5'-TCTTGTAAAACGTGAAGACTACCTG-3'<br>Reverse primer<br>5'-TCAGTGAAACTGGAGCAAGCA-3' | |
| COX6B2<br>(BioRP purify) | Forward primer<br>5'-AATGGTAGATGCCTCGCGTT-3'<br>Reverse primer<br>5'-TAGCTCAGGGAGTACCCGTT-3' | |
| List of primers for generation mutant of cDNA constructs | | |
| Myc-SDCBP_Δ1<br>(BioRP purify) | Forward primer<br>5'-ATTCGGTCGACCCAAGGGATTCGTGAAGTCATT-3'<br>Reverse primer<br>5'-GATCCCCGCGGCCGCTTAAACCTCAGGAATGGT-3' | |
| Myc-SDCBP_Δ2<br>(BioRP purify) | Forward primer<br>5'-TTCGGTCGACCTCTCTCTATCCATCTCTC-3'<br>Reverse primer<br>5'-CCCCGCGGCCGCTTACATGATTGTAATAGTAAC-3' | |
| Myc-SDCBP_Δ3<br>(BioRP purify) | Forward primer<br>5'-TTCGGTCGACCTCTCTCTATCCATCTCTC-3'<br>Reverse primer<br>5'-CCCCGCGGCCGCTTACCTGTCACGAATGGTCAT-3' | |

| REAGENT or RESOURCE | SOURCE | IDENTIFIER |
|---|---|---|
| Myc-SDCBP_Δ4 (BioRP purify) | Forward primer 5'-TTCGGTCGACCTCTCTCTATCCATCTCTC-3' 5'-CGTCCGTTCAAAGGGCTTAATTTCTGCTCTACG-3' Reverse primer 5'-GATCCCCGCGGCCGCTTAAACCTCAGGAATGGT-3' 5'-AGAGCAGAAATTAAGCCCTTTGAACGGACGATT-3' | |
| Myc-SDCBP_ΔPDZ1 (BioRP purify) | Forward primer 5'-GCCCGAATTCGGTCGACCCAAGGGATTCGTGAAGTC-3' Reverse primer 5'-GATCCCCGCGGCCGCTTAAAAGGGCCTGTCACGAAT-3' | |
| HA-FBXO22_Δ1 (BioRP purify) | Forward primer 5'-AAGCTTGGTACCATGTATCCTTACGACGTGCCTGACTACG CCGGTGGAGGTGGTAGCACCCATCGGAGCGTAACC-3' Reverse primer 5'-GGCCGCTCTAGACTCGAGTTTATCATTTAGATGACCCCAG-3' | |
| HA-FBXO22_Δ2 (BioRP purify) | Forward primer 5'-AAGCTTGGTACCATGTATCCTTACGACGTG-3' Reverse primer 5'-GGCCGCTCTAGAAGTTTAGATTCGGTGTCCACT-3' | |
| HA-HO-1_H25A (BioRP purify) | Forward primer 5'-GAGGCCACCAAGGAGGTGGCCACCCAGGCAGAGAAT-3' Reverse primer 5'-ATTCTCTGCCTGGGTGGCCACCTCCTTGGTGGCCTC-3' | |
| *List of oligonucleotides for short hairpin/small interfering RNA(s)* | | |
| *Human SDCBP shRNA plasmid kit (Locus ID6386)* | Origene | Cat#TL309594 |
| *Mouse SDCBP shRNA plasmid kit (Locus ID53378)* | Origene | Cat#TG512166 |
| *Human SDCBP siRNA Oligo Duplex (Locus ID6386)* | Origene | Cat#SR321723 |
| *Mouse SDCBP siRNA Oligo Duplex (Locus ID53378)* | Origene | Cat#SR426294 |
| *Human BACH1 siRNA* | Santa Cruz Biotechnology | Cat#sc-37064 |
| *Human NRF2 siRNA* | Santa Cruz Biotechnology | Cat#sc-37030 |
| *Human FBXO22 siRNA* | Santa Cruz Biotechnology | Cat#sc-90142 |
| *Human HMOX1 siRNA* | Santa Cruz Biotechnology | Cat#sc-35554 |
| Expression plasmids | | |
| pCMV-tag2B-N-Flag vector | Agilent | Cat#211172 |
| pUC19-hBACH1 cDNA | Sino Biological | Cat#HG12605-U |
| pCMV-Myc vector | Takara Bio USA, Inc. | Cat#631604 |
| pCMV3-Myc - SDCBP cDNA | Hwangbo et al (2015) | N/A |
| pCMV3-Myc - SDCBP_M1 cDNA | This study | N/A |
| pCMV3-Myc - SDCBP_M2 cDNA | This study | N/A |
| pCMV3-Myc - SDCBP_M3 cDNA | This study | N/A |
| pCMV3-Myc - SDCBP_M4 cDNA | This study | N/A |
| pCMV3-Myc- SDCBP_PDZ1 cDNA | This study | N/A |
| pCMV-N-HA vector | Takara Bio USA, Inc. | Cat#635690 |
| pCMV3-HA-RCBK1 (HOIL-1) cDNA | Sino Biological | Cat#HG19261-NY |
| pCMV3-HA-FBXO22 | Sino Biological | Cat#HG17164-NY |
| pCMV3-HA-FBXO22_M1 | This study | N/A |
| pCMV3-HA-FBXO22_M3 | This study | N/A |
| pCMV3-HA-RCBK1/HOIL1 | Sino Biological | Cat#HG19261-NY |
| pRK5-HA-Ubiquitin-WT | Addgene | Cat#17608 |
| pRK5-HA-Ubiquitin-K48 | Addgene | Cat#17605 |
| pCMV3-N-HA HMOX1 | Sino Biological | Cat#HG16608-NY |

| REAGENT or RESOURCE | SOURCE | IDENTIFIER |
|---|---|---|
| pCMV3-N-HA-HMOX1-H25A_Mut | This study | N/A |
| pcDNA3.1(+) | Thermo Scientific™ | Cat#V79020 |
| pcDNA3.1 FLAG NRF2 | Addgene | Cat#36971 |
| pCMV3-C-His-SKP1 cDNA | Sino Biological | Cat#HG14161-CH |
| pCMV3-N-FLAG-CUL1 cDNA | Sino Biological | Cat#HG17691-NF |
| pCMV3-SP-N-FLAG-PDL1/CD274 cDNA | Sino Biological | Cat#HG10084-NF |
| **Software and Algorithms** | | |
| ImageJ | ImageJ-NIH | https://imagej.net |
| Fiji | Fiji | https://fiji.sc |
| GraphPad ver10.0 | PraphPrism | https://www.graphpad.com |
| Venny ver2.1 | BioinfoGP | https://bioinfogp.cnb.csic.es |
| Seahorse XFe24 | Seahorse | https://www.agilent.com |
| QuPath ver0.4.4 | QuPath | https://qupath.github.io |
| ZEN lite ver 3.4 | Carl Zeiss Microscopy | https://www.zeiss.com |
| SnapGene™ ver 1.1.3 | Snapgene | https://www.snapgene.com |
| Gen5™ ver1.11.5 | Bio Tek Instruments Inc. | N/A |
| Olympus cellSens standard ver 2.3 | Olympus Corporation | N/A |
| Slideviewer ver 2.5 | 3DHistech Ltd. | https://www.3dhistech.com |
| FlowJo™ ver10.8.2 | Becton Dickinson & Company | https://www.flowjo.com |
| Eukaryotic Promoter Database (EPD) | Swiss Institute of Bioinformatics | https://epd.expasy.org/epd |

## Animal Experiments

All animal experiments were approved by the Institutional Animal Care and Use Committee at Kangwon National University, Gangwon-do, Republic of Korea (ethics approval numbers KW-220427-1 and KW-230613-1). Five-week-old female BALB/C and athymic BALB/c nude mice were purchased from DBL (Chungbug, Republic of Korea) and housed with 5 mice/cage in a 12:12-h light-dark cycle. Mice were randomly mammary fat-pad injected with stable SDCBP-KO MDA-MB-231 cells or 4T1 cells ($1 \times 10^5$ cells/mice). Every 3 days, tumor volumes were calculated as follows: Tumor volume = $0.5 \times$ length $\times$ width$^2$. For athymic nude ice experiment, the average tumor volume reached ~1000 mm$^3$, all mice were sacrificed by cervical dislocation. All tumor tissues were collected, fixed, and then processed for H&E or immunohisto-chemistry staining. For BALB/c mice experiment, Metformin (100 mg/kg; CAT#PHR1084, Sigma-Aldrich, USA) was administered daily to the mice via intraperitoneal injection. Adenoviral SDCBP shRNA treatments (CAT#shADV271-494, Vector Biolabs, USA) were provided via intra-tumor injection every 3 days. Tumor growths were monitored every 3 days. Mice survival was monitored daily, and bodyweights were regularly checked prior to treatment. When the average tumor volume of the vehicle group reached ~1000 mm$^3$, all tumor tissues and lungs were collected, fixed, and processed using H&E or immunohistochemistry staining.

## Cell culture

All cell lines were purchased from the American Type Culture Collection (ATCC, USA) as described in the Reagents and Tools table. The MDA-MB-231, MDA-MB-468, NCI-H1299 were maintained in Dulbecco's Modified Eagle medium (DMEM) (CAT#L0103-500, Biowest, USA) supplemented with 5% heat-inactivated fetal bovine serum (FBS) (CAT#S1480, Biowest, USA) and 1% penicillin/streptomycin (Pen/Step) (CAT#15140-122, Gibco, USA). The T47D, MCF-7, and 4T1 were maintained in the Roswell Park Memorial Institute 1640 (RPMI) (CAT#L0498-500, Biowest, USA) supplemented with 5% FBS and 1% Pen/Strep. The Hs578T were maintained in DMEM supplemented with 10% FBS and 1% Pen/Strep. The A549 and HEK293 were maintained in RPMI supplemented with 10% FBS and 1% Pen/Strep. For subculture, these cells were freshly washed with 1x PBS at pH 7.4 (CAT#10010-023, Gibco, USA) and detached by 0.05% trypsin-EDTA (CAT#25300-062, Gibco, USA). All cells were maintained in a humidified 5% $CO_2$ atmosphere at 37 °C.

## DNA constructs and transfection

Human or mouse wild-type/original cDNA constructs were purchased, and the specific details are provided in the Reagents and Tools table. The Myc-SDCBP, HA-FBXO22 mutant constructs were generated using rCutSmart™ Buffer kit (CAT#B6004S, New England BioLabs lnc., USA), according to the manufacturer's instructions. The point mutant construct for HA-HMOX1$^{H25A}$ was generated using a Muta-direct™ site-directed Mutagenesis Kit (CAT#15071, LiliF Diagnostics, Korea) following the manufacturer's protocol. All generated cloning plasmids and mutations were analyzed by sequencing services (Cosmo Genetech, Korea). The PCR primers are listed in the Reagents and Tools table. Cells were transfected with cDNAs or siRNAs using a Lipofectamine3000

transfection kit (CAT#L3000-015, Invitrogen, USA), according to the manufacturer's instructions.

## Generation of stable SDCBP knockout TNBC cells using CRISPR-Cas9

The CRISPOR software program (Zhang Lab, https://www.zlab.bio, Cambridge, USA) was used to designed and identified sgRNA, then cloned into pLentiCRISPRv2 according to a previously described protocol (Platt et al, 2014). SDCBP knockout seudo-lentiviruses were generated using PEI Max® Kit (CAT#24765-1000, Polysciences, USA) following the manufacturer's protocol. MDA-MB-231 cells were transduced with pseudo-lentivirus, then selected with 1 µg/mL puromycin (CAT#P8833, Sigma-Aldrich, USA) and 250 µg/mL hygromycin-B (CAT#H0654, Sigma-Aldrich, USA). SDCBP knockout single-cell clones were selected by using serial limiting-dilution. SDCBP protein expression was screened in each subclone using western blotting and RT-qPCR assays. Vector control cells were generated using an identical infection and cloning strategy with pLenti-CRISPRv2 empty vector pseudo-lentiviruses.

## RNA extraction and real-time quantitative polymerase chain reaction (RT-qPCR)

Total RNA was isolated using an RNA extraction kit (RNeasy® Mini Kit, 74106, Qiagen, Valencia, CA, USA), then reversely transcribed with Maxime RT-PreMix kit (CAT#25081, LiliF Diagnostics) to synthesize cDNAs, and subjected to. StepOne Real-Time PCR System machine (Applied Biosystems, Korea) using a Fast-qPCR-Master Mix SYBR kit (CatTQ1210, SMOBIO, Korea) following the manufacturer's protocols. The qPCR primers are listed in the Reagents and Tools table.

## Chromatin immunoprecipitation-quantitative polymerase chain reaction (ChIP-qPCR)

The ChIP-qPCR assay was performed using a Pierce™ agarose ChIP kit (CAT#26156, ThermoScientific, Korea) following the manufacturer's protocol. Briefly, MDA-MB-231 cells were seeded on 10 cm plates and fixed and cross-lined with 1% formaldehyde for 10 min and quenched with 1× glycine solution for 5 min at RT. The chromatins of interest were processed for cell pellet isolation, lysis, and MNase-digestion using Chromatin Prep Module Kit (CAT#26158, ThermoScientific, Korea) following the manufacturer's protocol. Chromatin immunoprecipitation was then conducted using the Pierce Chromatin Prep Module Kit (CAT#26158, ThermoScientific, Korea) following the manufacturer's protocol. Final purified DNA samples were subjected to RT-qPCR assay with the designed primers. The ChIP-qPCR primers are listed in the Reagents and Tools table.

## Western blotting assay

Western blotting assays were performed as previously described (Hwangbo et al, 2015). Briefly, tissue-extracted and cellular proteins were isolated with ice-cold RIPA buffer (CAT#R0278, Sigma-Aldrich, USA) and cOmplete-Protease inhibitor cocktail (CAT#11697498001, Roche-Diagnostics, Germany). Protein

concentrations were determined using a Bradford protein assay kit (CAT#B6916, Sigma-Aldrich, USA). The lysates of proteins interested were equally subjected to SDS-PAGE and transferred to PVDF membranes. Membranes were optionally blocked with PBS-T containing 5% skim milk or 5% BSA and probed with the indicated antibodies, as described in detail in the Reagents and Tools table. The immunoblots were visualized using an EZ-Western Solution Chemiluminescence Kit (CAT#DLS2305, DogenBIO, Korea) following the manufacturer's instructions.

## Immunoprecipitation (IP)

Immunoprecipitation assays were performed as previously described (Hwangbo et al, 2015). Cells were lysed in ice-cold RIPA buffer (CAT#R0278, Sigma-Aldrich, USA) and cOmplete-Protease inhibitor cocktail (CAT#11697498001, Roche-Diagnostics, Germany). Cell lysates were incubated with indicated antibodies at 4 °C for overnight. and then incubated with either protein A/G PLUS-agarose (CAT#sc2003, Santa Cruz, Korea) or Dynabead™ protein-G (CAT#10003D, Invitrogen, USA). The IP samples were washed four-times with ice-cold RIPA buffer and followed by Western Blotting analysis. To avoid unspecific bands, pre-clear steps of A/G PLUS-agarose, cross-linking agents BS$_3$ (#CAT21580, Thermo-Fisher Scientific, USA) and Veriblot kits (CAT#ab131369, CAT#131366, Abcam, USA) were used to specifically visualize protein–protein interactions of interest), according to the manufacturer's instructions.

## His-pulldown assay

Proteins were extracted from cultured cells as described for the western blotting assay above. The His-tagged protein complexes were isolated using Dynabeads™ His-Tag Isolation and Pulldown (CAT#10103D, Invitrogen, Carlsbad, CA, USA), according to the manufacturer's instructions. Eluted proteins were analyzed using a western blotting assay with the indicated antibodies as described above.

## In vivo ubiquitylation assay

Cells were transfected with the indicated cDNA constructs using a Lipofectamine3000 Transfection Kit (CAT#L3000-015, Invitrogen, Korea) and treated with MG132 proteasome inhibitor (CAT#C2211, Sigma-Aldrich, USA) for 6 h before harvesting. Protein lysates were extracted with ubiquitination-denaturing ice-cold lysis buffer containing protease inhibitor cocktail, then prepared for the IP assay. The polyubiquitin signals were visualized using a western blotting assay with the HA-Tag antibody (CAT#3724, Cell Signaling, USA) or ubiquitin antibody (CAT#sc8017, Santa Cruz, USA).

## In vitro ubiquitylation assay

In brief, HEK293 cells were transfected with the HA-FBXO22 cDNA construct using a Lipofectamine3000 Transfection Kit and treated with MG132 (CAT#C2211, Sigma-Aldrich, USA). The E3 ubiquitin ligase SCF$_{FBXO22}$ complex was isolated from the cultured cell lysate via immunoprecipitation of the HA-Tag antibody (CAT#3724, Cell Signaling, USA) using Dynabead™ protein-G

(CAT#10003D, Invitrogen, USA), according to manufacturer's instructions. The IP lysates were added with UbcH5A recombinant protein (CAT#ab269096, Abcam, USA), BACH1 recombinant protein (CAT#TP321628, OriGene, USA), and SDCBP recombinant protein (CAT#15640-H07E, Sino Biological, USA) following the indicated experimental designs, then performed *invitro ubiquitination* reactions by using Ubiquitylation Assay Kit (CAT#ab139647, Abcam, USA), followed by immunoblotting analysis according to manufacturer's instructions. The polyubiquitin signals were visualized using an ubiquitin-conjugated antibody, which was provided in the kit.

## Cell proliferation and cell viability assays

Cell viability was estimated by staining with the MTT thiazolyl blue reagent (CAT#M1415, Duchefa-Biochemie, Netherlands). Cells were then seeded at a density of $5 \times 10^3$ cells/well into 96-well plates. At the different time points, cells were stained with 0.5 mg/mL MTT thiazolyl blue reagent and maintained in a humidified 5% $CO_2$ atmosphere at 37 °C. After 4 h, isopropyl-alcohol (CAT#0005-00380, Duksan, Korea) was added into each well to measure viability values at 570 nm using an ELISA reader system (Agilent Technologies, Korea).

Cell proliferation was automatically estimated using the Countess 3 automated cell counter system (ThermoScientific, Korea), according to the manufacturer's protocol. Briefly, cells were detached by 0.05% Trypsin-EDTA (CAT#25300-062, Gibco, USA) and stained with trypan blue stain (CAT#T10282, Invitrogen, Korea). The proliferation rate was then determined as the cell numbers were normalized into percentage values for the indicated time points.

## Colony formation assay

The colony formation assay was performed as previously described (Kim et al, 2022). Briefly, cells were seeded at a density of $5 \times 10^2$ cells/well into a 6-well plate. After 2 weeks, all cells were washed twice and gently fixed with methanol for 15 min. All cells were stained with 0.05% w/v crystal violet (CAT#C0775, Sigma-Aldrich, USA) at RT for 30 min. Cell plates were then washed with distilled water and dried overnight. The next day, the colonies were counted, then analyzed using an ELISA reader (Agilent Technologies Inc.).

## Cell migration and wound-healing assays

Cell migration assays were performed as previously described (Kim et al, 2022; Hwangbo et al, 2015). Briefly, cells were seeded onto the fibronectin-coated SPLInsert™ insert chambers (CAT#37224, SPL Life Science, Korea). At the indicated time points, the migrated cells were fixed in methanol and stained with hematoxylin solution (CAT#ab220365, Abcam, USA) and eosin solution (CAT#E4382, Sigma-Aldrich, USA) and dried overnight at room temperature. The migration rate was estimated by counting the migrated cell number captured using an inverted phase-contrast microscope (Optical Microscope Olympus IX51/TH4-200, Japan).

For wound-healing assays, cells were seeded into 12-well plates and vertically scratched using sterilized yellow-plastic tips to generate a cell monolayer. At the indicated time points, the migration rate was estimated by measuring the remaining area of the scratched monolayer captured using the inverted phase-contrast microscope (Optical Microscope Olympus IX51/TH4-200; Japan).

## Free heme measurement

Free heme levels were estimated using the Heme Assay kit (CAT#ab272534, Abcam), according to the manufacturer's protocol. Briefly, cell lysates were isolated with ice-cold lysis buffer, then mixed with Heme assay reagent and incubated at RT for 15 min, followed by absorbance measurements at OD 400 nm. Free heme concentrations were calculated based on the OD value of the supplied calibrator. Distilled water was used as the blank, and all measurements were performed in triplicate.

## Mitochondrial membrane potential assay

Mitochondrial transmembrane potentials were measured using TMRE staining, as described in a previous study (Crowley et al, 2016). Briefly, cells were resuspended and incubated with culture media containing 150 nm TMRE (CAT#T669, Invitrogen) at RT for 5 min in the dark. Cell suspensions were subjected to BD FACSymphony™ A3 (Becton Dickinson, USA). The mitochondrial transmembrane potential was estimated basing on the TMRE intensity. The FCCP (CAT#04-513-0, ThermoFisher Scientific, USA) was used the positive control. The data were analyzed and visualized using the FlowJo™ software (Becton Dickinson, USA) within the licensed key belonging to Kangwon National University Central Laboratory (Chuncheon-si, Korea). Cells were also stained with 1 μM TMRE to visualize the mitochondrial membrane potential using Super Sensitive High Resolution Confocal Laser Scanning (Carl ZEISS, Germany). All confocal images were analyzed using ZEN Lite software (Carl ZEISS).

## Oxygen consumption rate measurement assay

The oxygen consumption rate was estimated using Seahorse XFe24V7 PS cell culture microplates (CAT#100777-004, Agilent, USA), according to the manufacturer's protocol. Briefly, cells were initially transfected with SDCBP siRNA oligo duplex (CAT#SR321723, OriGene, USA) and then resuspended at a density of $2 \times 10^4$ cells/well into XFe24V7 PS cell culture microplates. Cells were then cultured in growth medium in a humidified 5% $CO_2$ atmosphere at 37 °C. Then, 24 h after incubation, the oxygen consumption rate was automatically calculated using a mitochondrial stress test with the Seahorse XFe24 software (Agilent, USA). Further measurements were recorded after treatments with indicated inhibitors.

## Immunofluorescence staining and confocal imaging assays

Immunofluorescence staining assays were performed as previously described (Hwangbo et al, 2015). Briefly, cells were rinsed twice in ice-cold PBS, fixed with 4% paraformaldehyde (CAT#P6148, Sigma-Aldrich, USA), and permeabilized in 0.1% Triton™ X-100 (CAT#9036-16-5, Sigma-Aldrich, USA). Cells were incubated with indicated primary antibodies in a shaker at 4 °C overnight, followed by incubations with indicated secondary Alexa Fluor antibodies in a shaker at room temperature for 6 h. Cells were washed and stained

with 1 µg/mL DAPI (CAT#D9542, Sigma-Aldrich, USA) for 5 min in the dark. The stained cells were mounted and stored at 4 °C in the dark for further imaging applications. Immunofluorescence imaging was performed using the Time-lapse cell culture system microscope Olympus (IX81-ZDC; Olympus) or the Super Sensitive High Resolution Confocal Laser Scanning (Carl ZEISS). All immunofluorescence images were then analyzed using the ZEN Lite software (Carl ZEISS).

## Hematoxylin & eosin staining and immunohistochemical staining assays

For tumor staining, tumors were isolated and washed twice with ice-cold PBS, and the skin was completely removed. Tumor tissues were fixed in 4% paraformaldehyde (CAT#P6148, Sigma-Aldrich, USA) for 12 h, then in 30% sucrose for 12 h, prior to being embedded in the OCT compound. The entire cryomold was frozen in a dry ice bath and prepared for cryosection. Tumor tissues were cut into 6-µm sections for further immunohistochemical analyses. The sections were incubated with indicated antibodies, as listed in the Reagents and Tools table. After incubation with primary antibodies, the sections were washed twice with PBS-T, stained using a Dako Peroxidase/DAB+ Detection Kit (CAT#K5007, Agilent, USA) following the manufacturer's protocol, and then optionally counterstained using hematoxylin solution (CAT#ab220365, Abcam, USA). All sections were imaged by using the Digital Slide Scanner OCUS40 (Grundium, Finland), visualized using the Slideviewer software (3DHistech, Hungary), and then analyzed using the QuPath software (QuPath, https://qupath.github.io).

For tissue microarray staining, briefly, human TNBC tissue microarray sections (CAT#BR1301, CAT#BR1902, TissueArray-US Biomax, USA) were heated and deparaffinized. The sections were then incubated with SDCBP antibody (CAT#A5497, ABclonal, Korea), NDUFA4 antibody (Cat#PA5-51021, Invitrogen, Korea), or with BACH1 antibody (CAT#AF5776, R&D Systems, USA), and then stained using a Cell & Tissue Staining Kit (CAT#CTS005, CAT#CTS008, R&D Systems, USA), according to manufacturer's protocol. For picture representation, 3,3-diaminobenzidine was digitally separated via color deconvolution with ImageJ software. Picture representations were auto-processed using the TMA plugin and set to $200 \times 200$-pixel boxes at the size of 200 µm and 50 µm, respectively. The Spearman's test and correlation coefficient values were calculated as previously described Hwangbo et al, 2015.

## Analysis of TCGA data

GSE 142102 and GSE 103091 databases were downloaded from 'https://www.ncbi.nlm.nih.gov/geo/' website. GSE 142102 was based on GPL17692 (Affymetrix Human Gene 2.1 ST Array [transcript (gene) version]) and GSE 103091 was based on GPL570 (Affymetrix Human Genome U133 Plus 2.0 Array). GSE96508 database was downloaded from 'https://www.ncbi.nlm.nih.gov/geo/' website. For analysis, the probe name of each gene was searched from GSE data file to obtain expression values. Correlation between genes was analyzed through XY table in GraphPad prism® 10.01. The survival rate analysis of TNBC patients was gained from the Kaplan–Meier (KM) plotter website (https://kmplot.com/analysis/). For the survival rate of TNBC

patients, ER, PR, and HER2 were all fixed as negative and was analyzed by selecting the probe of a specific gene. Statistical significance was determined by two-tailed Student's t-test.

## RNA sequencing analysis

The experimental procedures were performed as previously described (Kim et al, 2022). Briefly, mRNAs were extracted from stable SDCBP knockout subclones using a RNeasy mini kit (CAT#74106, Qiagen, USA) according to the manufacturer's protocol. mRNA samples were sent to the Macrogen, Inc. (Seoul, Korea) for RNA sequencing. The data were then normalized and analyzed via bioinformatic approaches and graphed using Graph-Pad Prism software (GraphPad Software, USA).

## Graphical image

Graphical images and Synopsis images were created by using Biorender.com. All images were sublicensed for use in the *EMBO Journal* with agreement number WU274DKRCF.

## Quantification and statistical analyses

All statistical analyses were performed using GraphPad software (GraphPad Software). Data are expressed as the mean ± standard error of the mean (SEM). Statistically significant differences were measured with an unpaired two-tailed student's t-test when only two groups were compared; otherwise, a two-way or one-way analysis of variance (ANOVA). All experiments were repeated at least three times unless otherwise indicated. *P* values less than 0.05 were considered statistically significant.

# Data availability

The order number of NGS Analysis for RNA-seq data in this study is HN00172576. This is the private accession number provided by Macrogen Inc., Seoul, Republic of Korea. The online version of raw data can be accessed via the SRA database number PRJNA1084248. Source data are provided with this paper. Further information and requests for this resource should be directed to the Lead Contact.

The source data of this paper are collected in the following database record: biostudies:S-SCDT-10_1038-S44318-025-00440-1.

# Materials availability

Materials and reagents used in this study are listed in the Resources Table. Materials and reagents in this study from our laboratory are available upon request to the lead contact above.

# Peer review information

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

## Acknowledgements

This research was supported by a grant from the National Research Foundation (NRF) of Korea (NRF-2020R1A5A8019180).

## Author contributions

**Phi-Long Tran**: Conceptualization; Data curation; Formal analysis; Investigation; Methodology; Writing—original draft; Writing—review and editing. **Okhwa Kim**: Data curation; Formal analysis; Investigation; Methodology; Writing—review and editing. **Cheol Hwangbo**: Data curation; Formal analysis; Investigation; Methodology; Writing—review and editing. **Hyo-Jin Kim**: Data curation; Formal analysis; Investigation. **Young-Myeong Kim**: Formal analysis; Methodology;

Writing—review and editing. **Jeong-Hyung Lee**: Conceptualization; Supervision; Methodology; Writing—review and editing.

Source data underlying figure panels in this paper may have individual authorship assigned. Where available, figure panel/source data authorship is listed in the following database record: biostudies:S-SCDT-10_1038-S44318-025-00440-1.

## Disclosure and competing interests statement

The authors declare no competing interests.

ns licence, unless indicated otherwise in a credit line to the material. If material is not included in the article's Creative Commons licence and your intended use is not permitted by statutory regulation or exceeds the permitted use, you will need to obtain permission directly from the copyright holder. To view a copy of this licence, visit http://creativecommons.org/licenses/by/4.0/. Creative Commons Public Domain Dedication waiver http://creativecommons.org/public-domain/zero/1.0/ applies to the data associated with this article, unless otherwise stated in a credit line to the data, but does not extend to the graphical or creative elements of illustrations, charts, or figures. This waiver removes legal barriers to the re-use and mining of research data. According to standard scholarly practice, it is recommended to provide appropriate citation and attribution whenever technically possible.

© The Author(s) 2025

# Expanded View Figures

**Figure EV1. SDCBP regulates the expression of the BACH1 protein and its target genes in TNBC cells.**

(A) TCGA data analysis showing the correlation between *SDCBP* mRNA and *BACH1* mRNA expression in GSE142102 ($n = 226$) dataset of TNBC patients (Pearson correlation coefficient r = 0.3245, $P < 0.0001$). (B) TCGA data analysis showing the correlation between *SDCBP* mRNA and *BACH1* mRNA expression in GSE103091 ($n = 238$) dataset of TNBC patients (Pearson correlation coefficient r = 0.2120, $P < 0.001$). (C) Western blot showing SDCBP, BACH1, and HO-1 protein expression in MDA-MB-231, MDA-MB-468, Hs578T, MCF-7, and T47D cells. (D) The expression levels of SDCBP and BACH1 protein in Fig. EV1C were quantified using densitometry and normalized to the housekeeping protein α-tubulin ($n = 3$). (E) Real-time qPCR showing *SDCBP* and *BACH1* mRNA expression in MDA-MB-231, MDA-MB-468, Hs578T, MCF-7, and T47D cells ($n = 3$). Quantitative data were normalized to β-actin expression. (F) Western blot showing SDCBP and HO-1 protein expression in MDA-MB-231 cells transfected with scramble or BACH1 siRNA. (G) Left, western blot showing the protein expression of SDCBP in the scramble and in several SDCBP-KO MDA-MB-231 subclones generated using CRISPR-Cas9 system; Right, real-time qPCR showing the *SDCBP* mRNA expression in scramble and in SDCBP-KO MDA-MB-231 subclones ($n = 3$). (H) Real-time qPCR showing the mRNA expression of *BACH1* in MDA-MB-231 cells, in scramble and in SDCBP-KO MDA-MB-231 subclone#2 and subclone#12 ($n = 3$). (I) Immunofluorescence staining was used to visualize SDCBP (green color) and BACH1 (red color) in scramble and in SDCBP-KO MDA-MB-231 cells. DAPI (blue color) was used to stain the nucleus ($n = 3$; Representative confocal immunofluorescence images are shown. Scale bar = 20 μm. (J) Western blot showing BACH1 and HO-1 protein expression in 4T1 cells infected with scramble or adenoviral SDCBP shRNA. (K) Real-time qPCR showing the mRNA expression of BACH1-regulated antioxidant genes (*HMOX1, NQO1*, and *GLCL*) in 4T1 cells transfected with scramble or SDCBP siRNA ($n = 3$); mRNA expression of KEAP1 was the negative control. Data are expressed as the mean ± SEM and analyzed using one-way ANOVA (D, E, G, H) or two-way ANOVA (K). *P* values less than 0.05 were considered statistically significant. All experiments were repeated at least three times unless otherwise indicated.

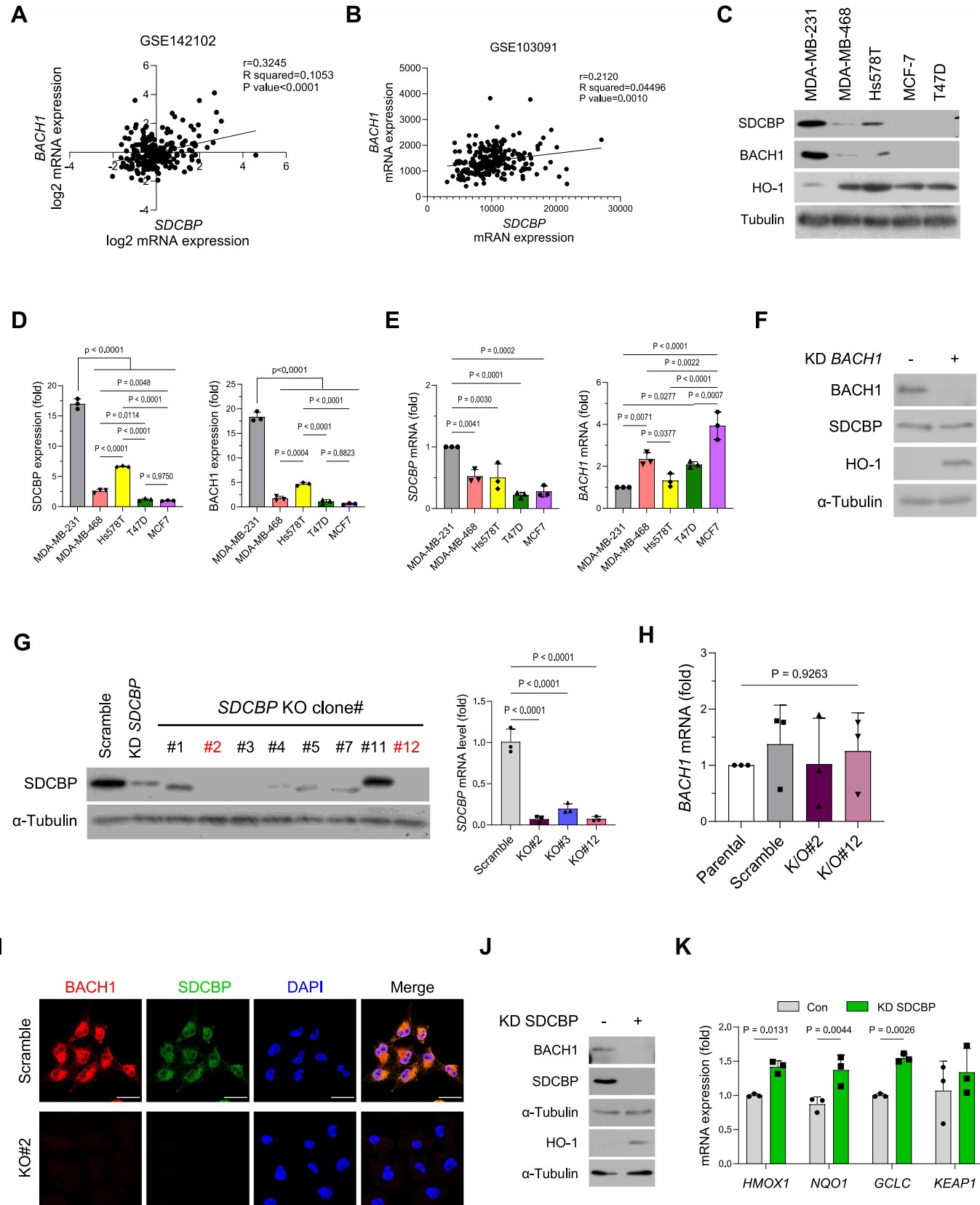

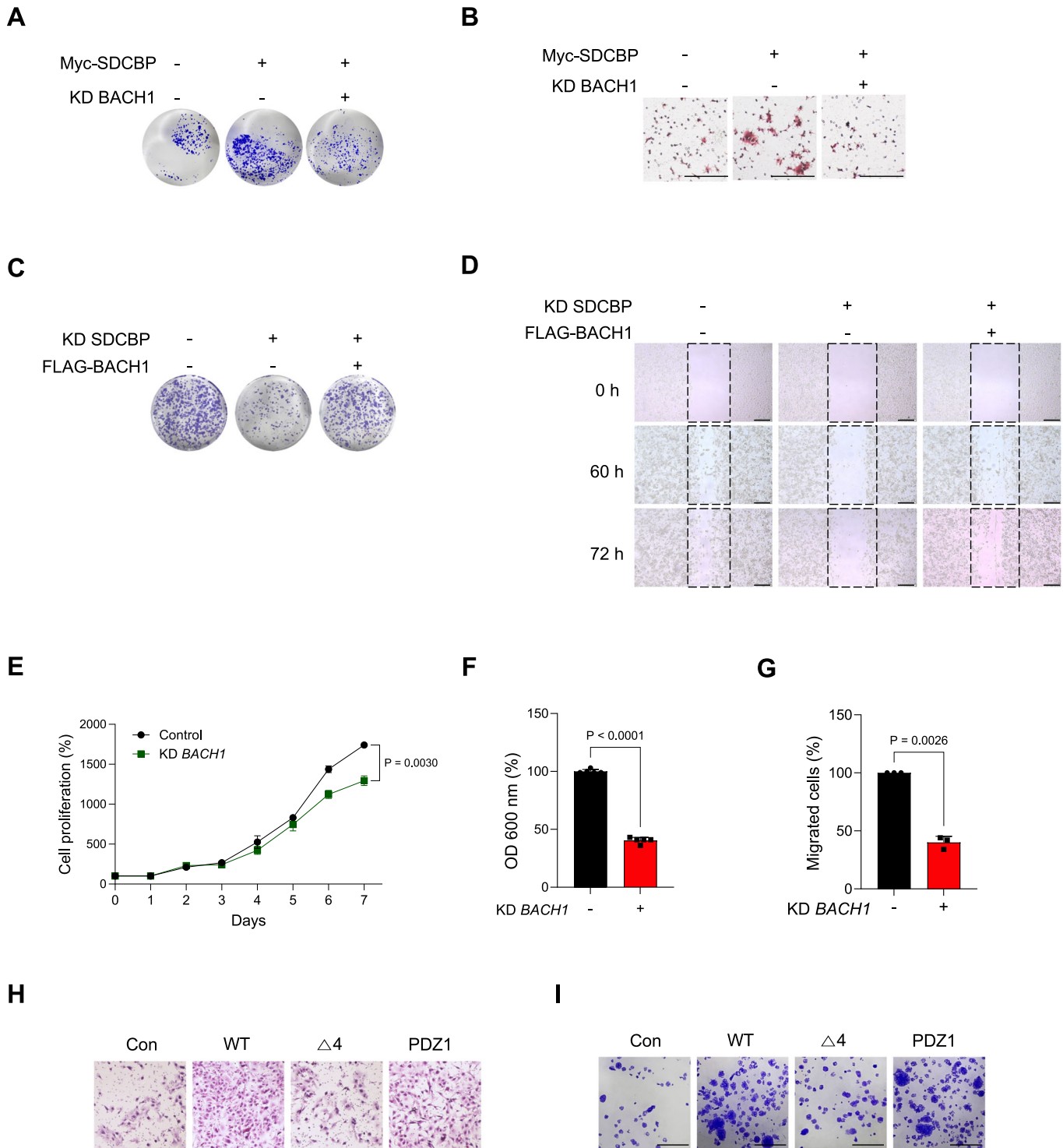

**Figure EV2.  SDCBP promotes tumor progression by upregulating BACH1 in TNBC cells.**

(A) Representative images of colony formation in Fig. 2B. (B) Representative images of the migrated cells in Fig. 2C. Scale bar = 200 μm. (C) Representative images of colony formation in Fig. 2F. (D) Representative images of wounding migration in Fig. 2G. Scale bar = 200 μm. (E) Cell proliferation of MDA-MB-231 cells transfected with scramble or BACH1 siRNA. Cell proliferation was estimated by an automatic cell counter at the indicated time points ($n = 3$). (F) Colony formation of MDA-MB-231 cells transfected with scramble or BACH1 siRNA. The clonogenic ability was assessed and quantified based on the absorbance at 600 nm and normalized to the control ($n = 3$). (G) Migration of MDA-MB-231 cells transfected with scramble or BACH1 siRNA. The number of migrated cells were counted and expressed as percentages ($n = 3$). (H) Representative images of the migrated cells in Fig. 2K. Scale bar = 500 μm. (I) Representative images of colony formation in Fig. 2L. Scale bar = 1000 μm. Data are expressed as the mean ± SEM and analyzed using two-tailed Student's $t$ test with Welch's correction (E–G). $P$ values less than 0.05 were considered statistically significant. All experiments were repeated at least three times unless otherwise indicated.

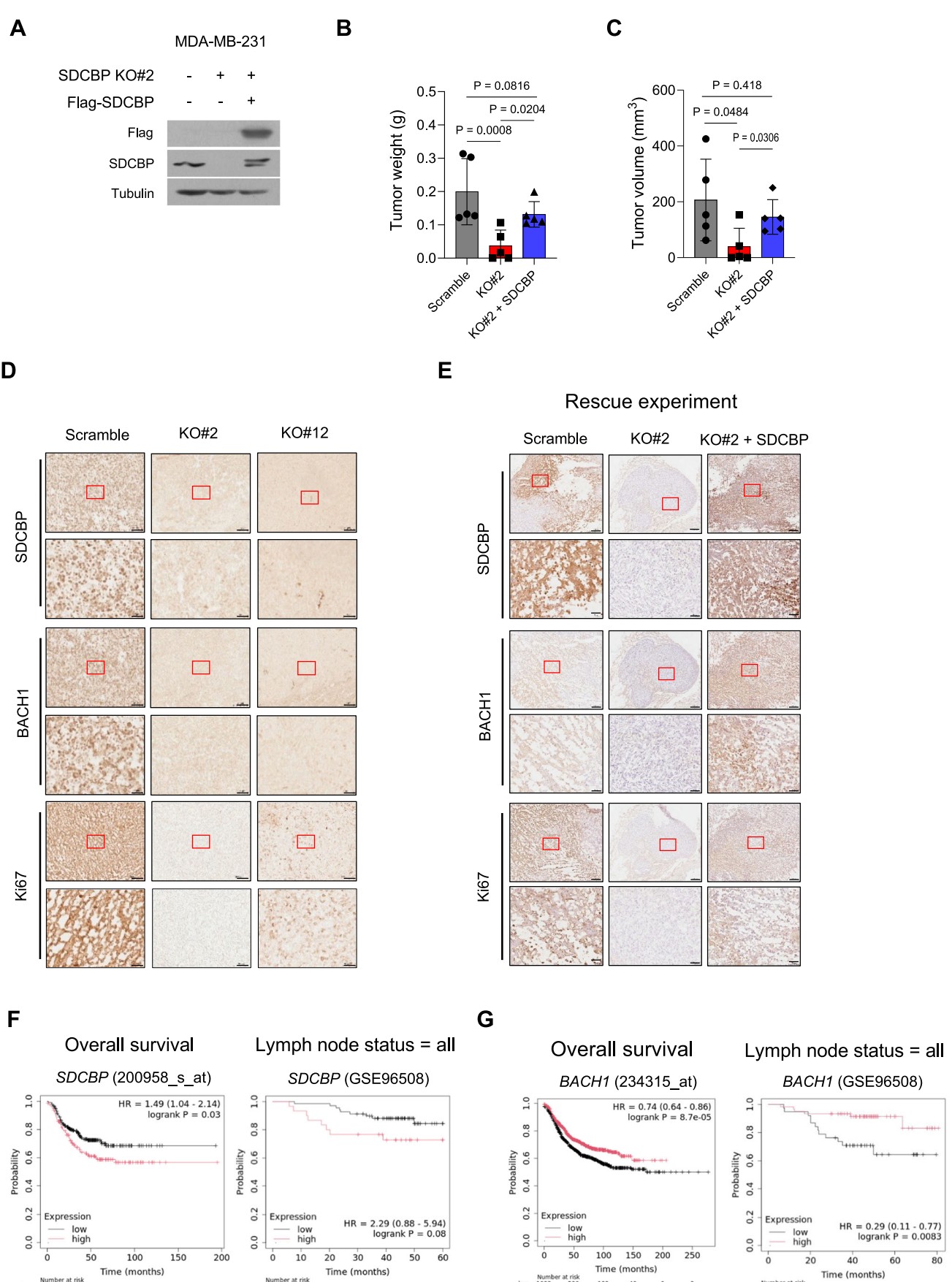

◀  **Figure EV3.  SDCBP promotes tumor growth by upregulating BACH1 and is associated with the survival of TNBC patients.**

(A) Western blot showing the expression of SDCBP and Flag-SDCBP in Fig. 2O. (B) Tumor weights in Fig. 2O ($n = 5$ mice/group). (C) Tumor volume in Fig. 2O ($n = 5$ mice/group). (D) Representative images of immunohistochemistry staining against SDCBP, BACH1, and Ki67 protein for xenografted tumors isolated from athymic BALB/c nude mice 6 weeks after mammary fat-pad injection of the scramble control or SDCBP-KO MDA-MB-231 cells ($1 \times 10^5$ cells/mouse; $n = 7$ mice/group). Representative images of IHC staining are shown. Scale bar $= 200\ \mu m$ (upper) and $50\ \mu m$ (lower), respectively. (E) Representative images of immunohistochemistry staining against SDCBP, BACH1, and Ki67 protein for xenografted tumors isolated from athymic BALB/c nude mice 25 days after mammary fat-pad injection of the scramble control, SDCBP-KO MDA-MB-231 cells, or SDCBP-KO MDA-MB-231 cells stably transfected with Flag-SDCBP ($1 \times 10^5$ cells/mouse; $n = 5$ mice/group). Scale bar $= 200\ \mu m$ (upper) and $50\ \mu m$ (lower), respectively. (F) TCGA data analysis showing association between *SDCBP* mRNA expression and overall survival ($n = 392$) and lymph node status ($n = 98$) of TNBC patients. (G) TCGA data analysis showing association between *BACH1* mRNA expression and overall survival ($n = 2032$) and lymph node status ($n = 98$) of TNBC patients. Data are expressed as the mean $\pm$ SEM and analyzed using two-way ANOVA (B) or two-tailed Student's $t$ test (C). $P$ values less than 0.05 were considered statistically significant. All experiments were repeated at least three times unless otherwise indicated.

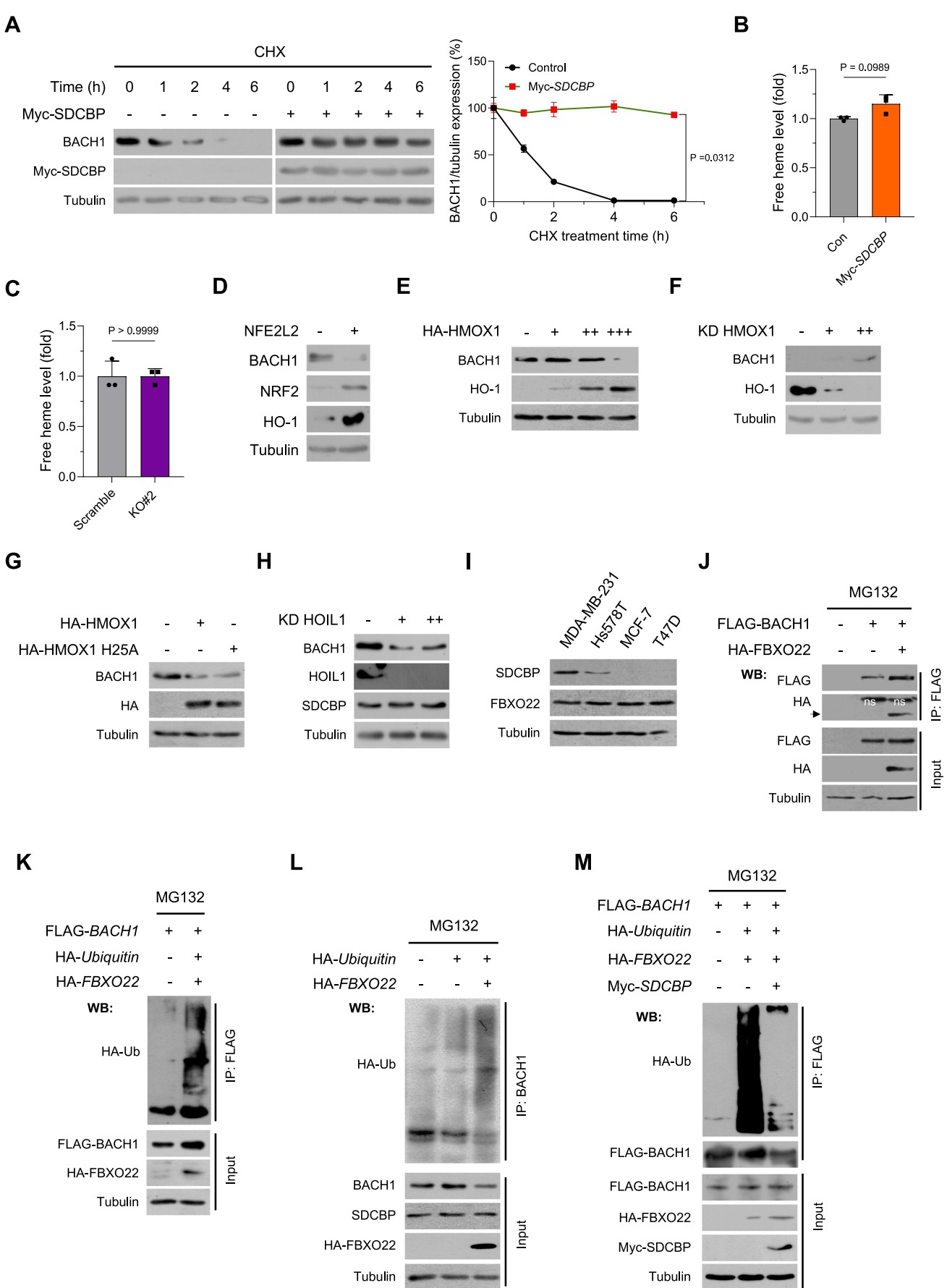

◀ **Figure EV4. SDCBP induces BACH1 stability by impairing FBXO22-mediated BACH1 polyubiquitination via an alternative Heme/HO-1-independent mechanism.**

(A) Left, Western blot showing BACH1 protein expression in MDA-MB-231 cells transfected with control vector or Myc-SDCBP-expressing vector in the presence of CHX protein synthesis inhibitor at various time points. Right, quantification of BACH1 protein levels using densitometry ($n = 3$). (B) Free heme level in Hs578T cells transfected with control vector or Myc-SDCBP-expressing vector ($n = 3$). (C) Free heme level in scramble and in SDCBP-KO MDA-MB-231 cells ($n = 3$). (D) Western blot showing BACH1 protein expression in MDA-MB-231 cells transfected with control vector or NRF2 (encoded by *NFE2L2*)-expressing vector. HO-1 protein expression was considered as the positive control. (E) Western blot showing BACH1 protein expression in MDA-MB-231 cells transfected with a control or a HO-1 (encoded by *HMOX1*)-expressing vector. (F) Western blot showing BACH1 protein expression in Hs578T cells transfected with scramble or HO-1 siRNA. (G) Western blot showing BACH1 protein expression in MDA-MB-231 cells transfected with the HO-1 or the catalytic inactive HO-1 mutant (H25A) plasmid. (H) Western blot showing BACH1 and SDCBP protein expression in MDA-MB-231 cells transfected with scramble or HOIL1 siRNA. (I) Western blot showing endogenous FBXO22 protein expression in several breast cancer cells. (J) Immunoprecipitation showing the interaction of BACH1 with FBXO22 in HEK293 cells transfected with the indicated plasmids. An arrow indicates the specific signal for HA-FBXO22. ns: none specific. (K) In vivo ubiquitylation assay showing the increase in the polyubiquitylation of BACH1 by FBXO22 overexpression in HEK293 cells transfected with the indicated plasmids. (L) In vivo ubiquitylation assay showing the increase in the polyubiquitylation of BACH1 by FBXO22 overexpression in MDA-MB-231 cells transfected with the indicated plasmids. (M) In vivo ubiquitylation assay showing the decrease in FBXO22-mediated polyubiquitylation of BACH1 by SDCBP overexpression in HEK293 cells transfected with the indicated plasmids. Data are expressed as the mean ± SEM and analyzed using two-tailed Student's *t* test with Welch's correction (A–C). All experiments were repeated at least three times unless otherwise indicated. *P* values less than 0.05 were considered statistically significant.

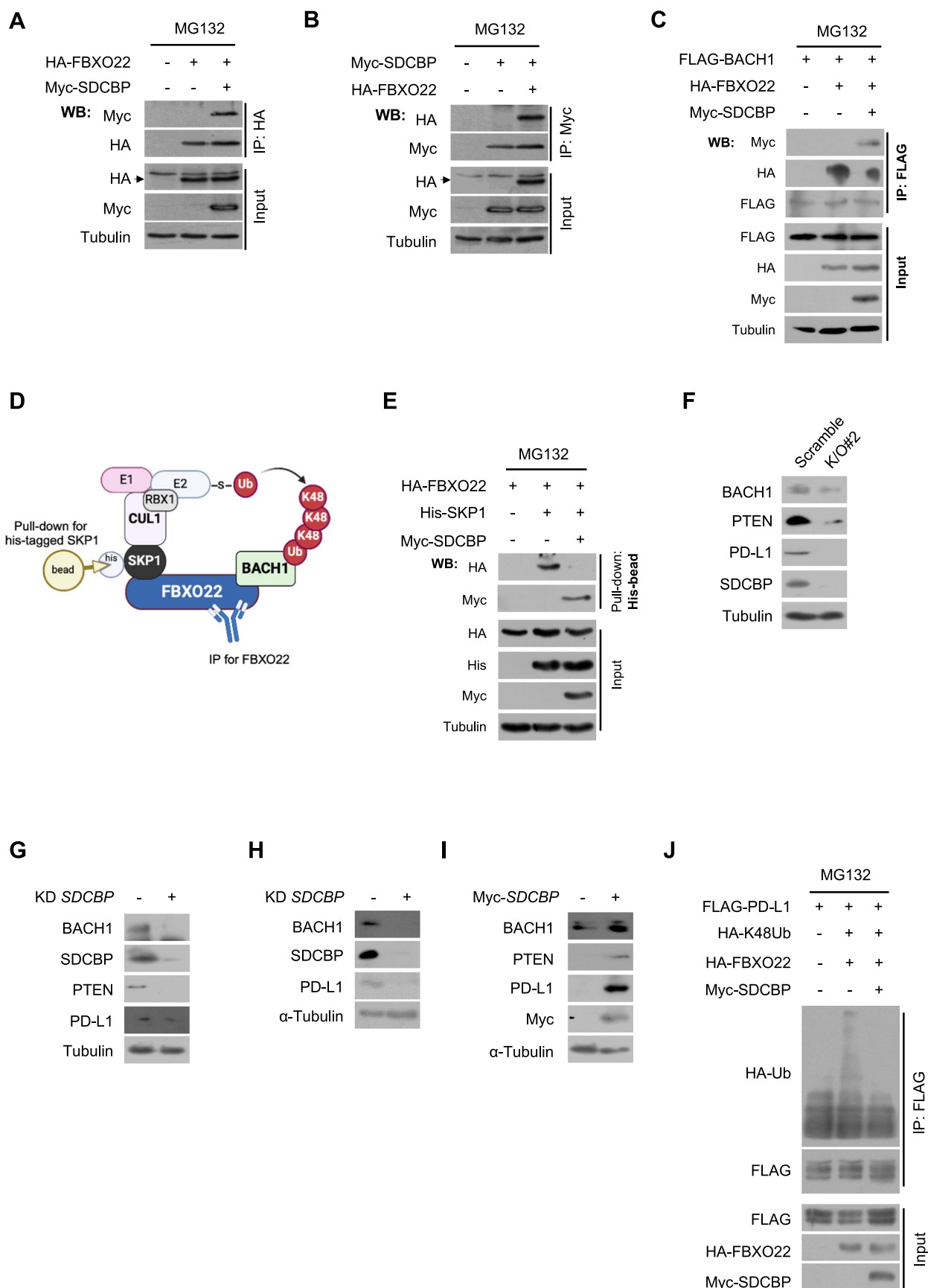

◀ **Figure EV5. SDCBP associates with FBXO22 and impairs SCF^FBXO22-targeted substrates for K48-linked degradative ubiquitination.**

(A) Immunoprecipitation showing the interaction of FBXO22 with SDCBP in HEK293 cells transfected with the indicated plasmids. An arrow indicates the specific signal for HA-FBXO22. (B) Immunoprecipitation showing the interaction of SDCBP with FBXO22 in HEK293 cells transfected with the indicated plasmids. An arrow indicates the specific signal for HA-FBXO22. (C) Co-immunoprecipitation showing the interaction of FBXO22 with BACH1 in HEK293 cells with or without SDCBP after the indicated transfections. (D) Schematic of experimental design to investigate the assembly of SCF^FBXO22–BACH1 complex via His Pull-down assay and endogenous IP assay in Fig. 4D–G. (E) His-pulldown assay showing the interaction of FBXO22 with SKP1 in HEK293 cells with control vector or Myc-SDCBP-expressing vector after the indicated transfections. See also Appendix Fig. S1A. (F) Western blot showing BACH1, PTEN, and PD-L1 protein expression in scramble and in SDCBP-KO MDA-MB-231 cells. (G) Western blot showing BACH1, PTEN, and PD-L1 protein expression in A549 cells transfected with scramble or SDCBP siRNA. (H) Western blot showing BACH1 and PD-L1 protein expression in NCI-H1299 cells transfected with scramble or SDCBP siRNA. (I) Western blot showing BACH1, PTEN, and PD-L1 protein expression in Hs578T cells transfected with control vector or Myc-SDCBP-expressing vector. (J) In vivo ubiquitylation assay showing the inhibitory effect of SDCBP on SCF^FBXO22-mediated K48-linked polyubiquitylation of BACH1 in HEK293 cells transfected with the indicated plasmids.

