## [Peer Review File · The EMBO Journal]

SDCBP/Syntenin-1 stabilizes BACH1 by disassembling the SCFF^{BXO22}-BACH1 complex in triple-negative breast cancer

Phi-Long Tran, Okhwa Kim, Cheol Hwangbo, Hyo-Jin Kim, Young-Myeong Kim⁴, and Jeong-Hyung Lee

Corresponding author: Jeong-Hyung Lee (jhlee36@kangwon.ac.kr)

Review Timeline:

Submission Date:	29th Jul 24
Editorial Decision:	8th Oct 24
Revision Received:	28th Jan 25
Editorial Decision:	6th Mar 25
Revision Received:	20th Mar 25
Accepted:	24th Mar 25

Editor: Daniel Klimmeck

Transaction Report:

Dear Dr Lee,

Thank you for the submission of your manuscript (EMBOJ-2024-118617) to The EMBO Journal as well for your patience with our response at this time of the year. The delay was due to protracted referee input and detailed discussion in the editorial team. Your study was assessed by three reviewers with expertise in cancer signaling and cellular metabolism, whose comments are enclosed below.

As you will see from the experts' reports, the referees acknowledge the analysis and potential interest of your results. However, they also express major concerns regarding completeness and detail of the findings, which need to be addressed thoroughly to make them supportive of publication in the EMBO Journal. The reviewers also raise a number of issues related to the data presentation, additional controls and improved methods annotation required, statistics applied and overall discussion of related literature, that would need to be conclusively addressed to achieve the level of robustness and clarity needed for The EMBO Journal.

Given the overall interest stated and broader angle of your findings, we are able to invite you to revise your manuscript experimentally to address the referees' comments. I need to stress though that we do require strong support from the referees on a revised version of the study in order to move on to publication of the work.

In light of the extensive experimentation requested, I would appreciate if you could contact me during the next weeks for exchange e.g. a video call to discuss your perspective on the comments and potential plan for revisions.

Please feel free to contact me if you have any questions or need further input on the referee comments.

When submitting your revised manuscript, please carefully review the instructions below.

Please feel free to approach me any time should you have additional questions related to this.

Thank you for the opportunity to consider your work for publication.

I look forward to your revision.

Kind regards,

Daniel Klimmeck

Daniel Klimmeck, PhD
Senior Editor
The EMBO Journal

Instruction for the preparation of your revised manuscript:

- 1) a .docx formatted version of the manuscript text (including legends for main figures, EV figures and tables). Please make sure that the changes are highlighted to be clearly visible.
- 2) individual production quality figure files as .eps, .tif, .jpg (one file per figure).
- 3) a .docx formatted letter INCLUDING the reviewers' reports and your detailed point-by-point response to their comments. As part of the EMBO Press transparent editorial process, the point-by-point response is part of the Review Process File (RPF), which will be published alongside your paper.
- 4) a complete author checklist, which you can download from our author guidelines ([https://wol-prod-cdn.literatumonline.com/pb-assets/embo-site/Author Checklist%20-%20EMBO%20J-1561436015657.xlsx](https://wol-prod-cdn.literatumonline.com/pb-assets/embo-site/Author%20Checklist%20-%20EMBO%20J-1561436015657.xlsx)). Please insert information in the checklist that is

also reflected in the manuscript. The completed author checklist will also be part of the RPF.

6) It is mandatory to include a 'Data Availability' section after the Materials and Methods. Before submitting your revision, primary datasets produced in this study need to be deposited in an appropriate public database, and the accession numbers and database listed under 'Data Availability'. Please remember to provide a reviewer password if the datasets are not yet public (see <https://www.embopress.org/page/journal/14602075/authorguide#datadeposition>).

7) Our journal encourages inclusion of *data citations in the reference list* to directly cite datasets that were re-used and obtained from public databases. Data citations in the article text are distinct from normal bibliographical citations and should directly link to the database records from which the data can be accessed. In the main text, data citations are formatted as follows: "Data ref: Smith et al, 2001" or "Data ref: NCBI Sequence Read Archive PRJNA342805, 2017". In the Reference list, data citations must be labeled with "[DATASET]". A data reference must provide the database name, accession number/identifiers and a resolvable link to the landing page from which the data can be accessed at the end of the reference. Further instructions are available at .

8) At EMBO Press we ask authors to provide source data for the main and EV figures. Our source data coordinator will contact you to discuss which figure panels we would need source data for and will also provide you with helpful tips on how to upload and organize the files.

Numerical data can be provided as individual .xls or .csv files (including a tab describing the data). For 'blots' or microscopy, uncropped images should be submitted (using a zip archive or a single pdf per main figure if multiple images need to be supplied for one panel). Additional information on source data and instruction on how to label the files are available at .

9) We replaced Supplementary Information with Expanded View (EV) Figures and Tables that are collapsible/expandable online (see examples in <https://www.embopress.org/doi/10.15252/emj.201695874>). A maximum of 5 EV Figures can be typeset. EV Figures should be cited as 'Figure EV1, Figure EV2" etc. in the text and their respective legends should be included in the main text after the legends of regular figures.

11) For data quantification: please specify the name of the statistical test used to generate error bars and P values, the number (n) of independent experiments (specify technical or biological replicates) underlying each data point and the test used to calculate p-values in each figure legend. The figure legends should contain a basic description of n, P and the test applied. Graphs must include a description of the bars and the error bars (s.d., s.e.m.).

We realize that it is difficult to revise to a specific deadline. In the interest of protecting the conceptual advance provided by the work, we recommend a revision within 3 months (6th Jan 2025). Please discuss the revision progress ahead of this time with the editor if you require more time to complete the revisions.

Referee #1:

The manuscript by Tran et al. describes the novel finding that SDCBP stabilizes the transcription factor BACH by targeting the ubiquitin conjugating complex SCF-Foxo22 disassembly. The resulting stabilization of BACH leads to the expression of its target genes which facilitates invasion of triple negative breast cancer cells. The authors also report that targeting SDCBP affects mitochondrial activity via the impact of BACH on genes of the electron transport chain and enhance to efficacy of metformin.

The strengths of the study are: figure 1 convincingly shows that expression of SDCBP affects BACH levels, figures 2 and 3 is a structure function type of analysis and show an effect of SDCBP on stability of BACH respectively and are overall convincing, and the figures showing impact of KD of SDCBP on tumor growth are also convincing.

Overall, the study offers plentiful of data supporting their conclusion. The major weakness is the logic of how the authors aimed at looking at SDCBP in the first place and data increasing the relevance of SDCBP/BACH axis in triple negative cancers using human data base analysis would elevate the impact.

Specific comments:

- 1- The introduction reads like a review and has too much details. It leaves the reader thinking that a lot is already known about BACH stability and the current question is merely incremental. Further, since BACH is over expressed in several cancer types the focus on triple negative breast cancer is not well justified.
- 2- The logic of focusing on SDCBP and BACH is not well articulated and seems to be justified only by the fact that both BACH and SDCBP are overexpressed in cancers. If it had been identified as a binding protein to BACH or in a screen of some kind, the justification would be more valid.
- 3- The IHC in figure 1a is of low quality as every cells seem to be positive and the number of samples tested is not mentioned.
- 4- Fig. 2 b-c, only quantification is shown, images of colonies and migration should be shown. Same comment applies to panel g.
- 5- KD HOIL does not only not increase BACH it actually decreases it according to their data in EV3H, how do the authors explain this?
- 6- The study offers nice data but it is limited to a few cell lines. Therefore, the data would benefit from validation in human dataset for instance can the authors define a gene signature that would be specific for SDCBP oe BACH oe cancer using the cell lines which oe or KD SDCBP and use that signature to determine overall survival, metastasis, lymph node status are affected? Is the signature sub-type specific?

Minor comments:

- Why use Myc antibody rather than SDCBP antibody in fig 1c since the SDCBP antibody in panel f seems much better?

Referee #2:

In this manuscript Phi-Long Tran and colleagues suggest that SDCBP stabilizes BACH1 by disassembling the FBXO22-BACH1 complex in TNBC cells. Interestingly, the authors point that this process is important to sensitize cells to metabolic inhibitors preventing tumor growth. In detail, BACH1 as a transcription factor facilitates tumor progression by supporting several processes and it is triggered by redox species changing levels. Interestingly, SDCBP stabilizes BACH1 preventing its degradation by interfering with the polyubiquitinating FBOX activity that targets it to degradation. BACH triggers pro tumorigenic functions and represses electron transport chain ECT genes. The authors mechanistically deciphered how molecularly these events occur and proposed that upon induced reduction of SDCBP BACH1 degrades, turns down pro tumorigenic genes, enhances ECT which can now be potentially targeted and used as an achilleas heel of breast cancer cells. The topic is timely, and triple negative breast cancer, albeit access to new antibody drug conjugates against TROP2A remains a large unmet medical need.

Overall, this is a well-executed experimental manuscript with used of several cells systems, a systematic and streamlined

hierarchical approach that is backed with interesting clinical and in vivo experimental observations. However, there are a few points that limit the enthusiasm.

Figure 1 and 6 rely on a series of immunohistochemistry staining, however it is unknown how the antibodies used have been validated. This is a must as in many occasions IHC is prone to non-specific reporting when the Abs used are not of clinical grade. Please take advantage of cell pellets from cell lines with loss-of-function for the proteins of interest to confirm the staining specificity (ie. BACH1 and SDCBP).

Several of the interaction experiments are completed using overexpressing conditions. This is acceptable to an extent, but prone to artifacts. The endogenous protein remains and this may alter stoichiometries. IPs in endogenous conditions are preferred or at least key experiments validated using mild overexpression in the endogenous gene depleted conditions.

The authors used a series of human and mouse well-established cell lines to strengthen and generalize their findings. These are reasonably used in figure 1 and so on, yet when exploring the role of SDCBP some of the initially used cell lines are excluded and HEK293 (which are not breast cancer cell lines) are used to validate the findings. This is odd, reads awkward and points to cherry picking cellular systems that fit the hypothesis. HEK293 are not a valid model and Hs578T cells are expected in the context. Please address.

Similarly, why lung cancer cells such as A549 are used? If it is as a positive control or any other cause it should be explicit in the text, otherwise it is puzzling.

The authors suggest that MDA231, Hs578T, MCF7 and T47D cells express similar levels of FBXO22. They go on to show that in MDA231 this is the mediator of BACH1 regulation. What happens in ER+ BCa cell lines, MCF7 and T47D if BACH1 is expressed? Is the mechanism specific of ER- only?

Figure 2 M to O provides interesting in vivo data. However, there are some inconsistencies. Each KO has a very different phenotype. Figure 1 J shows that the KO achieved is similar yet the differences observed may be due to off target effects. Please address by using rescue experiment.

Figure 2M-O and 6E- F lead the authors to the interpretation that there are differences in metastasis. The proposed effect in both cases in primary tumors is clear and unequivocal, however the implication in metastasis seems too generous. It is well-known and established for decades in the field that the size of the tumor matters in terms of metastasis as there is more or less dissemination of cancer cells. Thus, in this case the observation is not due to an intrinsic property of the cancer cells but rather a consequence of the tumor size. For the latter size-matched tumors and not time-matched should have been used. Please modify the text to avoid overstating the implications of the observations, which in terms of primary tumor is remarkable and sufficient for the importance of the finding.

Referee #3:

The manuscript by Tran et al., suggest that SDCBP regulates BACH1 stability with concomitant effect on ETC (increase) and metastatic gene expression (decrease) in triple negative breast cancer. This is a new concept, which adds to the list of known BACH1 regulators, posing the principal question of - when. Authors need to establish the physiological conditions under which the proposed new mechanism takes place. Along these lines, most if not all data is based on ectopic expression of genes; the expression of endogenous proteins needs to be better demonstrated. Data from patient specimens support authors hypothesis yet experiments that directly establish the relevance of the proposed regulation are lacking. Since there are three ubiquitin ligases reported for BACH1, authors are expected to demonstrate when SDCBP, and no other mechanism prevails.

Additional points are listed below

Figure 2 - panel D (and quite a few other panels in this ms) lacks control in which Flag BACH1 is expressed in the absence of KD-SDCBP. Panel E requires a western blot to reveal level of BACH1 expression (again, a control missing in other experiments). Panel E lacks control indicated for panel D. What is the difference between K/O #2 and #12 (panels M,N,O)? Data demonstrating the relative expression of SDCBP and BACH1 under each of these KD is required.

Figure 3 - was the pulse chase data presented in graphs adjacent to the westerns calculated based on the relative expression of BACH1 to Tubulin (Panels B, C)? Controls for expression of BACH1 in the absence of UBCH5a and FBXO22 IC (alone and combination) is lacking (panel F). Controls are also lacking in panel G. quality of ubiquitination blots shown in panels H and I is poor.

Figure 4 - Controls (like those noted above) are lacking in panels D, E, F. quality of ubiquitination blot presented in panel G is poor. Controls without MG132 should be included in panels G,H,I.

Figure 5 - level of endogenous protein should be presented. Experiments should include both KD and OE. Panel F need to be explained in context of data shown in Fig 2M.

Figure 6 - level of BACH1 expression in lung mets shown in panels E and F need to be provided (i.e. IHC).

Response to Reviewers' Comments

Response to Referee #1:

The manuscript by Tran et al. describes the novel finding that SDCBP stabilizes the transcription factor BACH by targeting the ubiquitin conjugating complex SCF-Foxo22 disassembly. The resulting stabilization of BACH leads to the expression of its target genes which facilitates invasion of triple negative breast cancer cells. The authors also report that targeting SDCBP affects mitochondrial activity via the impact of BACH on genes of the electron transport chain and enhance to efficacy of metformin.

The strengths of the study are: figure 1 convincingly shows that expression of SDCBP affects BACH levels, figures 2 and 3 is a structure function type of analysis and show an effect of SDCBP on stability of BACH respectively and are overall convincing, and the figures showing impact of KD of SDCBP on tumor growth are also convincing.

Overall, the study offers plentiful of data supporting their conclusion. The major weakness is the logic of how the authors aimed at looking at SDCBP in the first place and data increasing the relevance of SDCBP/BACH axis in triple negative cancers using human data base analysis would elevate the impact.

RE]

As suggested, we added human TCGA database analysis showing the relevance of the SDCBP/BACH axis in triple-negative cancers and described accordingly in this revised version (Please See Fig EV1A, EV1B, EV3F, EV3G, EV7A, and EV7B).

Specific comments:

1] The introduction reads like a review and has too much details. It leaves the reader thinking that a lot is already known about BACH stability and the current question is merely incremental. Further, since BACH is over expressed in several cancer types the focus on triple negative breast cancer is not well justified.

RE]

As suggested, we edited the "Introduction" section in this revised version.

2] The logic of focusing on SDCBP and BACH is not well articulated and seems to be justified only by the fact that both BACH and SDCBP are overexpressed in cancers. If it had been identified as a binding protein to BACH or in a screen of some kind, the justification would be more valid.

RE]

As described in the introduction, both BACH1 and SDCBP have been reported as biomarkers for TNBC. The rationale for the present study to focus on SDCBP and BACH1 is as follows;

- 1) NRF2 activation is known to stabilize BACH1 protein via inducing HO-1 for heme degradation (Lignitto et al, 2019). In our preliminary studies, the expression level of HO-1, the rate-limiting enzyme for heme degradation, did not correlate with that of BACH1 in TNBC cell lines (Fig EV1C and EV1D).
- 2) The ectopic expression of NRF2 or HO-1 in MDA-MB-231 cells did not increase BACH1 expression (Fig EV4D and EV4E) and knockdown of HO-1 in Hs578T cells, which express high HO-1 levels, did not decrease BACH1 expression (Fig EV4F). Moreover, blocking the catalytic function of HO-1 by H25A mutation did not change the impact of HO-1 on BACH1 expression (Fig EV4G), suggesting that BACH1 may be stabilized by a heme/HO-1-independent mechanism in TNBC cells.
- 3) The ectopic expression of SDCBP in TNBC cells induced BACH1 expression but reduced the expression of BACH1-repressed target genes including *HMOX1* and *NQO1* (Fig 1C–1E). In contrast, knockdown or knockout of SDCBP reduced BACH1 expression but increased the expression of BACH1-repressed target genes (Fig 1F–K and Fig EV1F, EV1H, EV1I–K). Notably, re-expression of SDCBP into SDCBP-knockout TNBC cells restored BACH1 expression (Fig 1L), suggesting that SDCBP may regulate BACH1 expression.

Therefore, we investigated the molecular mechanism for SDCBP-dependent BACH1 stabilization in TNBC cells and evaluated the role of the SDCBP–BACH1 axis in tumor progression and metastasis, both *in vitro* and *in vivo*.

3] The IHC in figure 1a is of low quality as every cells seem to be positive and the number of samples tested is not mentioned.

REJ

As suggested, we edited Fig 1A and described the number of TNBC tissues in Figure Legend of Fig 1B in the revised version.

4] Fig. 2 b-c, only quantification is shown, images of colonies and migration should be shown. Same comment applies to panel g.

REJ

As suggested, we added the original images of Fig. 2B, C, F, and G to Fig EV2A, EV2B, EV2C, and EV2D in the revised version.

5] KD HOIL does not only not increase BACH1 it actually decreases it according to their data in EV3H, how do the authors explain this?

REJ

HOIL, also known as RCBK1, is dependent on heme molecules to facilitate the degradation of BACH1 via the ubiquitin-proteasome system in murine erythroleukemia cells (Zenke-Kawasaki Y, et al. Mol Cell Biol. 2007. PMID: 17682061). Similar to a previous study showing that HOIL does not interact with BACH1 and does not induce BACH1 degradation in human lung cancer cells (Lignitto *et al*, Cell. 2019 Jul 11;178(2):316-329.e18), our results show that KD of HOIL1 in TNBC cells did not increase BACH1 expression, suggesting that HOIL1 may not function as an E3 for BACH1 degradation in TNBC cells. Furthermore, we also found that FBXO22 function as the major E3 ligase for SDCBP-mediated BACH1 stabilization in a heme-independent mechanism. Thus, the possibility that HOIL1 is involved in SDCBP-mediated BACH1 stabilization was excluded. As the reviewer pointed out, KD of HOIL1 in TNBC cells actually decreased BACH1 expression. In the present study, we focused on SDCBP-induced BACH1 stabilization and its underlying mechanism in TNBC cells. However, further studies are needed to completely understand the impact of HOIL on BACH1 stability in TNBC cells.

6] The study offers nice data but it is limited to a few cell lines. Therefore, the data would benefit from validation in human dataset for instance can the authors define a gene

signature that would be specific for SDCBP or BACH or cancer using the cell lines which or KD SDCBP and use that signature to determine overall survival, metastasis, lymph node status are affected? Is the signature sub-type specific?

RE]

We greatly appreciate your helpful comments. As suggested, we added human TCGA database analysis showing the relevance of the SDCBP/BACH axis in TNBC and described accordingly in this revised version (Please See Fig EV1A, EV1B, EV3F, EV3G, EV7A, and EV7B).

High SDCBP protein expression has been reported to be associated with increased tumour size ($r = 0.421$, $P < 0.001$), presence of lymph node metastasis ($r = 0.221$, $P = 0.044$) and poor overall survival ($P = 0.01$) and recurrence-free survival ($P = 0.007$) in TNBC patients (Breast Cancer Res Treat. 2018 Sep;171(2):345-357. doi: 10.1007/s10549-018-4833-8.) In consistent with this report, our TCGA analysis reveals that *SDCBP* mRNA expression may be associated with the aggressiveness of TNBC patients.

In addition, we added TCGA analysis data showing that *NDUFA4* and *COX6B2* mRNA expression may associated with favorable outcomes of TNBC patients (Fig EV7A and EV7B). We also analyzed *NDUFA4* protein expression in TNBC tumor tissues using a human TNBC tissue microarray, and added this data in Fig 5G and 5H and described accordingly in this revised version.

Minor comments:

Why use Myc antibody rather than SDCBP antibody in fig 1c since the SDCBP antibody in panel f seems much better?

RE]

In panel Fig 1c, western blotting was performed using Myc antibody to determine the expression level of exogenous Myc-SDCBP. In panel Fig 1f, we knocked down endogenous SDCBP, thus performed a western blotting using SDCBP antibody to confirm the knockdown efficiency of endogenous SDCBP.

Response to Referee #2:

In this manuscript Phi-Long Tran and colleagues suggest that SDCBP stabilizes BACH1 by disassembling the FBXO22-BACH1 complex in TNBC cells. Interestingly, the authors point that this process is important to sensitize cells to metabolic inhibitors preventing tumor growth. In detail, BACH1 as a transcription factor facilitates tumor progression by supporting several processes and it is triggered by redox species changing levels. Interestingly, SDCBP stabilizes BACH1 preventing its degradation by interfering with the polyubiquitinating FBOX activity that targets it to degradation. BACH triggers pro tumorigenic functions and represses electron transport chain ECT genes. The authors mechanistically deciphered how molecularly these events occur and proposed that upon induced reduction of SDCBP BACH1 degrades, turns down pro tumorigenic genes, enhances ECT which can now be potentially targeted and used as an achilleas heel of breast cancer cells. The topic is timely, and triple negative breast cancer, albeit access to new antibody drug conjugates against TROP2A remains a large unmet medical need. Overall, this is a well-executed experimental manuscript with used of several cells systems, a systematic and streamlined hierarchical approached that is backed with interesting clinical and in vivo experimental observations. However, there are a few points that limit the enthusiasm.

1] Figure 1 and 6 rely on a series of immunohistochemistry staining, however it is unknown how the antibodies used have been validated. This is a must as in many occasion IHC is prone to non-specific reporting when the Abs used are not of clinical grade. Please take advantage of cell pellets from cell lines with loss-of-function for the proteins of interest to confirm the staining specificity (ie. BACH1 and SDCBP).

REJ

We understand the reviewer's concern. The antibodies utilized for immunohistochemistry staining are high-quality and come highly recommended by the manufacturer. However, for detecting SDCBP and BACH1 in Western Blotting experiments, these antibodies are not the optimal choice. We took great care in selecting and optimizing the antibody options for all our experiments. Of course, the specificity of the IHC antibodies was confirmed by detections of SDCBP and BACH1 proteins in SDCBP- or BACH1-knockout MDA-MB-231 cells using western blotting. The western blot results were as follows:

2] Several of the interaction experiments are completed using overexpressing conditions. This is acceptable to an extent, but prone to artifacts. The endogenous protein remains and this may alter stoichiometries. IPs in endogenous conditions are preferred or at least key experiments validated using mild overexpression in the endogenous gene depleted conditions. The authors used a series of human and mouse well-established cell lines to strengthen and generalized their findings. These are reasonably used in figure 1 and so on, yet when exploring the role of SDCBP some of the initially used cell lines are excluded and HEK293 (which are not breast cancer cells lines) are used to validate the findings. This is odd, reads awkward and points to cherry picking cellular systems that fit the hypothesis. HEK293 are not a valid model and Hs578T cells are expected in the context. Please address.

RE]

We understand the reviewer's concern.

The cell line HEK293, derived from Human Embryonic Kidney, is widely accepted and frequently used in research to initially investigate human protein interactions. We understand that further confirmation of these interactions in breast cancer cell lines is necessary to support the hypothesis. In line with feedback from reviewers, we have validated these interactions using both exogenous IPs with HEK293 cells (e.g., Fig 3G, Fig 4C – 4F) and endogenous IPs with TNBCs cells, including Hs57T8 and MDA-MB-231, (e.g., Fig 4A, Fig 4G – 4I, EV5G). These combinational approaches aim to strengthen and generalize our novel molecular mechanism findings.

3] Similarly, why lung cancer cells such as A549 are used? If it is as a positive control or any other cause it should be explicit in the text, otherwise it is puzzling.

RE]

We understand the reviewer's concern. BACH1 has been reported as a substrate of the SCF^{FBXO22} complex in lung cancer cell lines, including A549 and H1299 (Weil *et al*, 2019; Lignitto *et al*, 2019). In the present study, we investigated the role of SDCBP in stabilizing BACH1 protein by disrupting the assembly of SCF^{FBXO22}-BACH1 complex in TNBC cells. To gain a deeper understanding of how SDCBP regulates the SCF^{FBXO22} complex, we further explored the effect of SDCBP on the expression of SCF^{FBXO22}-substrates in both human lung cancer and TNBC cell lines. Our data suggests that SDCBP could be a promising target for globally regulating the activity of SCF^{FBXO22} E3 ligase in both lung cancer and TNBC cells (Fig EV5H – EV4K). Importantly, A549, H1299, MDA-MB-231, and Hs578T are human cancer cell lines that highly express SDCBP, which strengthens and broadens the applicability of our novel molecular mechanism finding. To prevent confusing the readers, we organized these data into EV figures and explain these explanations in the manuscript's discussion.

4] The authors suggest that MDA231, Hs578T, MCF7 and T47D cells express similar levels of FBXO22. They go on to show that in MD231 this is the mediator of BACH1 regulation. What happens in ER+ BCa cells lines, MCF7 and T47D if BACH1 is expressed? Is the mechanism specific of ER- only?

RE]

As shown in Fig EV1C, ER-positive breast cancer cell lines, such as T47D and MCF-7 cells, barely express SDCBP and BACH1. Therefore, we focused on the TNBC cells which highly express in SDCBP and BACH1. Our aim is to investigate how the SDCBP-BACH1 axis contributes to tumor progression and to elucidate the mechanism of the SDCBP-BACH1 axis in TNBC cells. While the reviewer's comment is very interesting, but we think that this study is beyond the scope of the current study. However, we plan to study whether SDCBP can stabilize BACH1 protein in ER-positive human breast cancer.

5] Figure 2 M to O provides interesting in vivo data. However, there are some inconsistencies. Each KO has a very different phenotype. Figure 1 J shows that the KO achieve is similar yet the differences observe may be due to off target effects. Please address by using rescue experiment.

RE]

We agree with the reviewer's comment. As suggested, we performed rescue experiments in athymic Balb/c mouse. Re-expression of SDCBP into SDCBP KO MDA-MB-231 cells (KO#2) rescued tumor growth of MDA-MB-231 cells in athymic Balb/c mice, similar to the scramble control group. We added this new data in the revised version and described it accordingly (Please see Fig 2O, Fig EV2A, EV3B, EV3C, and EV3E).

6] Figure 2M-O and 6E-F lead the authors to the interpretation that there are differences in metastasis. The proposed effect in both cases in primary tumors is clear and unequivocal, however the implication in metastasis seems too generous. It is well-known and establish for decades in the field that the size of the tumor matters in terms of metastasis as there is more or less dissemination of cancer cells. Thus, in this case the observation is not due to an intrinsic property of the cancer cells but rather a consequence of the tumor size. For the latter size-matched tumors and not time-matched should have been used. Please modify the text to avoid overstating the implications of the observations, which in terms of primary tumor is remarkable and sufficient for the importance of the finding.

RE]

We fully agree with the reviewer's comment. As suggested, we deleted the data related metastasis and modified this revised version of manuscript.

Response to Referee #3:

The manuscript by Tran et al., suggest that SDCBP regulates BACH1 stability with concomitant effect on ETC (increase) and metastatic gene expression (decrease) in triple negative breast cancer. This is a new concept, which adds to the list of known BACH1 regulators, posing the principal question of - when. Authors need to establish the physiological conditions under which the proposed new mechanism takes place. Along these lines, most if not all data is based on ectopic expression of genes; the expression of endogenous proteins needs to be better demonstrated. Data from patient specimens support authors hypothesis yet experiments that directly establish the relevance of the proposed regulation are lacking. Since there are three ubiquitin ligases reported for BACH1, authors are expected to demonstrate when SDCBP, and no other mechanism prevails.

RE]

Thank you for your valuable comments. Our study aims to discover the new molecular mechanisms by which SDCBP regulates BACH1 stability by impairing the assembly of the SCF^{FBXO22} complex, and demonstrate that SDCBP-BACH1 axis could be a potential target to suppress TNBC tumor progression. Furthermore, our findings propose a novel combinational strategy to target by SDCBP to improve metformin sensitivity in TNBC. We have validated the mechanism by both overexpression (exogenous) and knockout or knockdown (endogenous) systems in cell lines and mouse models.

To detect the SCFFBXO22-BACH1 complex, we have validated these interactions using both exogenous IPs with HEK293 cells (e.g., Fig 3G, Fig 4C – 4F) and endogenous IPs with TNBCs cells, including Hs57T8 and MDA-MB-231, (e.g., Fig 4A, Fig 4G – 4I, EV5G). These combinational approaches aim to strengthen and generalize our novel molecular mechanism findings. To demonstrate the relevance of our proposed mechanism in TNBC patients, the revised version included data from patient specimens (TNBC tumor tissue microarray and TCGA data analysis; please see Fig 1A, Fig 1B, Fig EV1A, Fig EV1B, Fig EV3F, Fig EV3G, Fig 5G, Fig 5H, and Fig EV7).

To clarify the functional mechanism of SDCBP in regulating BACH1 protein stability, we used both SDCBP overexpression and SDCBP knockdown or knockout systems. For knockdown by siRNA, the knockdown efficiency of endogenous SDCBP or BACH1 was confirmed by Western blotting using SDCBP or BACH1 antibody. For overexpression by transfection of Myc-SDCBP, Flag-SDCBP, or Flag-BACH1, the transfection efficiency of exogenous SDCBP or BACH1 was confirmed by Western blotting using Myc antibody or Flag antibody.

Additional points are listed below

1] Figure 2 - panel D (and quiet few other panels in this ms) lacks control in which Flag BACH1 is expressed in the absence of KD-SDCBP. Panel E require a western blot to reveal level of BACH1 expression (again, a control missing in other experiments). Panel E lacks control indicated for panel D. What is the difference between K/O #2 and #12 (panels M, N, O)? Data demonstrating the relative expression of SDCBP and BACH1 under each of these KD is required.

RE]

Our study aims to discover the new molecular mechanisms by which SDCBP regulates BACH1 stability by impairing the assembly of the SCF^{FBXO22} complex, and demonstrate that

SDCBP-BACH1 axis could be a potential target to suppress TNBC tumor progression.

In Fig 1, we found that SDCBP induced BACH1 protein expression. Thus, we confirmed that SDCBP mediates tumor progression by inducing BACH1 protein expression in Fig 2. For this, we used two TNBC cells; MDA-MB-231 cells express high levels of SDCBP and BACH1 (Fig EV1C, Nature. 2019 Apr;568(7751):254-258;) and Hs578T cells express low level of SDCBP and BACH1 (Fig EV1C). Thus, to provide the evidence that SDCBP mediates cell proliferation and migration via BACH1, Hs578T cells expressed Myc-SDCBP or co-expressed it with siSNA of BACH1 (Fig 2A, 2B, and 2C), and MDA-MB-231 cells knocked down SDCBP by siRNA or co-expressed it with Flag-BACH1 (Fig 2D, 2E, 2F and 2G). Because MDA-MB-231 cells express high levels of SDCBP and BACH1 (Nature. 2019 Apr;568(7751):254-258), we did not include a set overexpressing FLAG-BACH1 in the absence of KD-SDCBP. However, we included a set overexpressing FLAG-BACH1 in the absence of KD-SDCBP in Fig 2D and 2E as suggested in the revised manuscript.

To select knockout clones using CRISPR/Cas9 technology, a single cell should be isolated. As shown in Fig EV1G and EV1H, we used CRISPR/Cas9 technology to knockout SDCBP in MAD-MB-231 cells. To select SDCBP knockout MDA-MB-231 clones, we performed limiting dilution and selected with SDCBP knockout clones, KO#2 and KO#12, for further investigation. The expression data SDCBP and BACH1 were shown in Fig 1J. Furthermore, Fig EV3D showed SDCBP and BACH1 expression data by IHC in scrambled, KO#2, and KO#12 xenograft tumors in the revised version.

3] Figure 3 - was the pulse chase data presented in graphs adjacent to the westerns calculated based on the relative expression of BACH1 to Tubulin (Panels B, C)? Controls for expression of BACH1 in the absence of UBCH5a and FBXO22 IC (alone and combination) is lacking (panel F). Controls are also lacking in panel G. quality of ubiquitination blots shown in panels H and I is poor.

Re]

We edited these errors (Panels B, C) in the revised version.

We understand the reviewer's comments. Fig 3F and 3I show data from *in vitro* ubiquitination assay. In the *in vitro* system, ubiquitination of the substrate cannot occur without the addition of E1, E2, or E3. As indicated, recombinant Ubch5a was used as E2, recombinant BACH1 was used as SCF-FBXO22 substrate, and FBXO22

immunoprecipitates were used as E3. In Fig 3F and 3I, we showed that FBXO22 ubiquitinates BACH1 *in vitro* and recombinant SDCBP may inhibit this FBXO22-mediated BACH1 ubiquitination *in vitro*. Overall, these *in vitro* data support our conclusion that SDCBP may stabilize BACH1 by inhibiting FBXO22-mediated BACH1 ubiquitination and proteosomal degradation. To do this experiment again, we would have to buy the recombinant proteins again, which is costly and burdensome. Please understand this situation.

As suggested, we repeated ubiquitination assays and added these new data in the revised version (Fig. 3G and 3H).

4] Figure 4 - Controls (like those noted above) are lacking in panels D, E, F. quality of ubiquitination blot presented in panel G is poor. Controls without MG132 should be included in panels G, H, I.

RE]

In Fig. 4D, 4E, 4F, we examined the effect of SDCBP on the interaction between BACH1 and FBXO22 in HEK293 cells by the transfection of indicated plasmids. We believe that all blots necessary to interpret the interaction were shown. We examined the effect of SDCBP on the formation of the SKP1-CUL1-FBXO22 complex FBXO22 in HEK293 cells by the transfection of indicated plasmids using pull-down assays or IPs.

In Fig 4G, we determined the effect of SDCBP KD on the SCF-FBXO22-BACH1 complex formation by immunoprecipitation assays. We successfully detected an increase in SKP1-CUL1-FBXO22-BACH1 complex in the presence of MG-132 upon knockdown of SDCBP in MDA-MB-231 cells. Because knockdown of SDCBP in MDA-MB-231 cells increased the ubiquitination of BACH1 (Fig 3H), BACH1 blot could become smear band. As reviewer 2 asked us to provide the endogenous SKP1-CUL1-FBXO22-BACH1 interaction, we present His pull-down assay data in Fig 4G, demonstrating that SDCBP knockout increases the formation of the SKP1-CUL1-FBXO22-BACH1 complex. The original Fig 4G has been moved to Fig EV5G.

In the absence of a proteasome inhibitor such as MG-132, it is not easy to detect the interaction between the E3s and substrates for proteasomal degradation, and the ubiquitination of a substrate. Thus, researchers typically detect the interaction between E3s and a substrate, and the ubiquitination of a substrates in the presence of MG-132. SCF-FBXO22-substrate complex is a multi-protein complex; it is not easy to detect this complex.

Previous studies have detected the interaction between SCF-FBXO22 and the substrate for proteasomal degradation, or SCF-FBXO22-mediated ubiquitination of a substrate in the presence of MG-132 (Nat Commun. 2016 Feb 12;7:10574. doi: 10.1038/ncomms10574.; Proc Natl Acad Sci U S A. 2019 Jun 11;116(24):11754-11763. doi: 10.1073/pnas.1820990116.; J Clin Invest. 2018 Dec 3;128(12):5603-5619. doi: 10.1172/JCI121679.; Proc Natl Acad Sci U S A. 2021 Nov 23;118(47):e2112674118. doi: 10.1073/pnas.2112674118.; Cell. 2024 Dec 26;187(26):7568-7584.e22. doi: 10.1016/j.cell.2024.10.012.) Moreover, the half-life of BACH1 in TNBC cells was less than 1.5 h (Fig. 3B, 3C, and Fig EV3A), indicating that BACH1 is rapidly degraded via the proteasome pathway. This means that the interaction of SCF-FBXO22 with BACH1 could not be detected in the absence of proteasome inhibitor MG-132. As suggested, we performed this experiment again to detect the SCF-FBXO22 BACH1 complex without MG-132. However, we were not able to detect the SCF-FBXO22 BACH1 complex as follows. We believe that these additional data are not necessary to reach any conclusions.

5] Figure 5 - level of endogenous protein should be presented. Experiments should include both KD and OE. Panel F need to be explained in context of data shown in Fig 2M.

RE]

The levels of endogenous BACH1 and SDCBP were shown in Fig 1J. MDA-MB-231 cells express high levels of SDCBP and BACH1 (Nature. 2019 Apr;568(7751):254-258; Fig EV1C). Our results showed that SDCBP increased BACH1 stability by impairing the assembly of the SCF^{FBXO22} complex. We found that several ETC genes are regulated by SDCBP-BACH1 axis. Thus, in Fig 5A to 5E, we determined whether SDCBP KO- or KD-mediated upregulation of ETC genes, including COXB62 and NDUFA4, is downregulated by re-expression of BACH1.

To monitor the transfection of FLAG-BACH1 plasmid, we detected FLAG-BACH1.

We performed several rescue experiments by re-expression of SDCBP in SDCBP KO MDA-MB-231 cells and added these new data in the revised version (Fig 5B and Fig EV6F).

Fig 2M showed the tumor growth in a xenograft model, and data in Fig 5F showed the effect of SDCBP knockdown on the cytotoxicity of Metformin in MDA-MB-231 cells. These data indicate that SDCBP inhibition may suppressed tumor growth of TNBC cells in vivo and increase the sensitivity of Metformin against TNBC cells.

6] Figure 6 - level of BACH1 expression in lung mets shown in panels E and F need to be provided (i.e. IHC).

Re]

Reviewer 2 requested that the metastasis data be removed from in the revised version. However, However, IHC of lung metastases was performed and the following results were obtained;

Dear Dr Lee,

Thank you for submitting your revised manuscript (EMBOJ-2024-118617R) to The EMBO Journal, as well for your patience with our response. Your amended study was sent back to the three referees for their scientific re-evaluation, and we have received detailed comments from all of them, which I enclose below. As you will see, the experts state that the work has been substantially enhanced by the revisions and they are now broadly in favour of publication, pending minor revision.

Thus, we are pleased to inform you that your manuscript has been accepted in principle for publication in The EMBO Journal.

Please carefully consider the remaining minor points raised by adding appropriate antibody controls or alternatively removing the respective data from Figures 1 and 6, toning down related claims. Please also revisit the introduction, and consistency of protein nomenclature used by amending and adjusting the manuscript text where appropriate.

Further, we now need you to take care of a number of issues related to formatting and data presentation as detailed below, which should be addressed at re-submission.

Please contact me at any time if you have additional questions related to below points.

Thank you for giving us the chance to consider your manuscript for The EMBO Journal. I look forward to your final revision.

Again, please contact me at any time if you need any help or have further questions.

Best regards,

Daniel Klimmeck

>> Please limit the keywords for your study to maximally five.

>> Provide a completed Author Checklist.

>> Author Contributions: Remove the author contributions information from the manuscript text. Note that CRediT has replaced the traditional author contributions section as of now because it offers a systematic machine-readable author contributions format that allows for more effective research assessment. and use the free text boxes beneath each contributing author's name to add specific details on the author's contribution.

More information is available in our guide to authors.
<https://www.embopress.org/page/journal/14602075/authorguide>

>> Correct order of manuscript sections: Abstract / Keywords / Introduction / Results / Discussion / Methods / Data Availability / Acknowledgements / Disclosure and competing interests statement // References / Figure legends / Tables and their legends /

Expanded View Figure legends

>> Data availability section: should be moved to the end of Methods; "Materials Availability" should be removed.

>> Appendix: remove the abbreviations list from the appendix file, and introduce abbreviations in the running text at first appearance. Limit the number of EV figures to maximally five. Add the others supplemental figures as "Appendix Figure S1, S2 etc" to the Appendix file. Please add a table of contents, including page numbers. adjust callouts in figure legends and manuscript text accordingly.

>> Figure callouts: Please ensure that the figure panel Fig 2H is called out in sequential order.

>>Synopsis image: needs to be uploaded as separate high-resolution image file; add a paragraph to the Methods along the following format: "Graphics: (some of the... OR Figure #... OR synopsis) Graphics were created with BioRender.com."

>> Source data: provide additional numerical source data for Fig 5B; files need to be organized in folders and uploaded as one (zipped) file per figure.

>> Consider additional changes and comments from our production team as indicated below:

- Figure legends:

1. Please indicate what */ **/ ***/ **** represents; if this represents p value(s), please indicate the statistical test used and where appropriate the exact p value in the legend(s) of figure(s) 1E, H, I, K; 2B, C, E, F, G, I, K, L, M, N; 3B, C; 5A, B, D, E, F; EV1 D, E, G, K; EV2 E-G; EV3 B, C; EV4 A, EV6 E; EV8 A-D
2. Please note that the exact p values are not provided in the legends of figures 1B, EV1 A; EV7 A
3. Please indicate the statistical test used for data analysis in the legends of figures 1D, G; EV1 A, B, H; EV4 B, C; EV6 A, EV7 A
4. Please note that information related to n is missing in the legends of figures 1D, E, G, H, I, K; 2I, N; 3B, C; 5A, B, D, E, F; 6A, B, C, E, F; EV1 D, E, G, H, K; EV3 B, C; EV4 A-C; EV6 A, D, E; EV8 G.
5. Please note that the error bars are not defined in the legends of figures 1D, E, G, H, I, K; 2B, C, E, F, G, I, K, L, M, N, O; 3B, C; 5A, B, D, E, F; 6A, B, C, D, E, F; EV1 D, E, G, H, K; EV2 E-G; EV3 B, C; EV4 A-C; EV6 D, E; EV8 A, B, C, D, G.
6. Please note that scale bar and its definition are missing for figures EV2 B, D; EV2 L.

Referee #1:

The revised manuscript has improved the manuscript and the authors have been responsive to previous critiques.

Remaining minor concerns:

- The logic of focusing on SDCBP in the introduction remains unclear (lines 88-93). While it is understood that the study was designed to understand the mechanism of HO-independent stabilization of BACH, why focusing on SDCBP specifically other than the fact that it is also over expressed in TNBC? There are thousand of proteins that are over expressed in TNBC, why did the authors choose to focus on SDCBP?
- The authors use SYN in the model on figure 7 and in panel J-K in Fig. EV2 but every where else they use SDCBP. The authors should use only one designation for the protein throughout.

Referee #2:

The revised version of the manuscript by Phi-Long et al has improved from the revision process. Indeed, the authors have addressed some of my concerns, providing experimental data and analyses to substantiate some of their claims or causal relationships among the findings.

Of importance, the experiments regarding the control of potential off targets of the knockdowns, which is now provided is very relevant, similarly some of the IHC controls in western blot analyses is relevant. The reorganization of some of the data, extension of some of the immunoprecipitations using the right cellular have substantiated some of the authors claims. Similarly, the extension of the clinical validation strengthens the manuscript.

However, some technical raised points have been scarcely dealt with. IHC antibodies have not been validated using fixed cell pellets from cells with and without the target proteins, but western blots, which do not fulfill the same goal. Similarly, the rationale why HEKs were used is a poor explanation. The fact that others used an inappropriate system due to lack of models does not justify keeping using them. The new MDA-231 experiments are reassuring.

In summary, the manuscript has improved through the revision process. The scientific claims are overall strengthened.

Referee #3:

The revised manuscript addresses most but not all points raised by the reviewers. Some figures remain without requested controls, a point requiring either to remove the data or redo the experiments. Accordingly, conclusions drawn need to be toned down.

Response to the reviewers' comments

Response to the Reviewer#1

- The logic of focusing on SDCBP in the introduction remains unclear (lines 88-93). While it is understood that the study was designed to understand the mechanism of HO-independent stabilization of BACH, why focusing on SDCBP specifically other than the fact that it is also over expressed in TNBC? There are thousands of proteins that are over expressed in TNBC, why did the authors choose to focus on SDCBP?

RE]

As suggested by reviewer, the introduction was added and edited to clarify the reasons to focus our study on SDCBP in TNBC at lines 88-96 in this revised version.

As described in the previous response, we found that SDCBP and BACH1 were co-expressed in TNBC tissues, and that the expression level of SDCBP correlates well with that of BACH1 in TNBC cell lines, independent of HO-1. Therefore, we investigated the molecular mechanism by which SDCBP regulates BACH1 expression in TNBC cells, and found that SDCBP induces the stabilization BACH1 protein through the formation of SCFF^{BXO22}-BACH1 complex in the end.

-The authors use SYN in the model on figure 7 and in panel J-K in Fig. EV1 but everywhere else they use SDCBP. The authors should use only one designation for the protein throughout.

RE]

The Figure 7 and Figure J-K in Fig EV1 were replaced to unify the protein name to SDCBP rather than SYN-1 in this revised version. Thank you for pointing out this error.

Response to the Reviewer#2

- The revised version of the manuscript by Phi-Long et al has improved from the revision process. Indeed, the authors have addressed some of my concerns, providing experimental data and analyses to substantiate some of their claims or causal relationships among the findings. Of importance, the experiments regarding the control of potential off targets of the knockdowns, which is now provided is very relevant, similarly some of the IHC controls in western blot analyses is relevant. The reorganization of some of the data, extension of some of the immunoprecipitations using the right cellular have substantiated some of the authors claims. Similarly, the extension of the clinical validation strengthens the manuscript. However, some technical raised points have been scarcely dealt with. IHC antibodies have not been validated using fixed cell pellets from cells with and without the target proteins, but western blots, which do not fulfill the same goal.

RE]

We understand reviewer's concerns. We took great care in selecting and optimizing the IHC antibody options for all our experiments. Clear and specific signal of BACH1 was observed in IHC images using high-quality scanner (Digital Slide Scanner OCUS40 - Grundium Ltd., Tampere, Finland) and further confirmed by the rescue experiment in Figure EV3E. In addition, the specificity of the used antibodies was double-confirmed with the specific bands in the western blot data. Notably, the SDCBP (2C12, Abnova, Cat#H00006386-M01) and BACH1 (F-9, Santa Cruz Biotechnology, Cat#sc-271211) antibodies used in this study for IHC were also used for IHC in other groups (Oncogene 2019;38, 6781–6793. <https://doi.org/10.1038/s41388-019-0920-5>; J Clin Invest. 2023;133(20):e169671. <https://doi.org/10.1172/JCI169671>.

-Similarly, the rationale why HEKs were used is a poor explanation. The fact that others used an inappropriate system due to lack of models does not justify keeping using them. The new MDA-231 experiments are reassuring. In summary, the manuscript has improved through the revision process. The scientific claims are overall strengthened.

RE]

The cell line HEK293, derived from Human Embryonic Kidney, is widely accepted and frequently used in research to initially investigate human protein interactions (*Richard et al, (2017) Assembly and Function of Heterotypic Ubiquitin Chains in Cell-Cycle and Protein Quality Control, Cell, <https://doi.org/10.1016/j.cell.2017.09.040>*).

In line with feedback from reviewer#2, we have validated these interactions using both exogenous IPs with HEK293 cells (e.g., Fig 4C and 4D) and **endogenous IPs with TNBCs cells**, including **Hs57T8 and MDA-MB-231, (e.g., Fig 4A, Fig 4E – 4H).** We also determined the effect of SDCBP on BACH1 stability and SCF^{FBXO22}-mediated BACH1 ubiquitination in TNBC cells (e.g., Fig 3A – 3E, and Fig 3G, 3H). These combinational approaches aim to strengthen and generalize our novel molecular mechanism findings.

Response to the Reviewer#3

-The revised manuscript addresses most but not all points raised by the reviewers. Some figures remain without requested controls, a point requiring either to remove the data or redo the experiments. Accordingly, conclusions drawn need to be toned down.

RE]

About Figure 2D and 2E: In the first-round review, the reviewer concerned about the control in which FLAG-BACH1 is expressed. We appreciate the reviewer comment and update a new version of Figure 2D and 2E by adding the control as requested.

About Figure 3F and 3G: In the first-round review, *the reviewer also pointed out that there is no control lane for BACH1 in Fig 3F and 3G in which UBCH5a and/or FBXO22 IC is absent. Fig 3F is an *in vitro* ubiquitination assay and Fig 3G is *in vivo* ubiquitination assay data.* In the *in vitro* ubiquitination assay system, ubiquitination reaction of the substrate does not occur without the complete addition of E1, E2, and E3. The lane for BACH1 in which UBCH5a (playing role for E2) or only FBXO22-IC (playing role for E3)

would be not ubiquitinated due to lacking full E1-E2-E3 complex formation. In this manuscript, we mainly focused on the inhibitory effect of SDCBP on SCF^{FBXO22}-mediated degradative ubiquitination of BACH1. We showed that SCF^{FBXO22} induced BACH1 ubiquitination in an *in vitro* system (Fig. 3F) and that the ubiquitination was inhibited by SDCBP in an *in vitro* assay system (Fig. 3I). To confirm these *in vitro* data, we performed in *in vivo* ubiquitination assays by overexpression or knockdown of SDCBP (Fig 3G and Fig 3H). Overall, these *in vitro* and *in vivo* data support our conclusion. In Fig 3G, we performed *in vivo* (in cell system) ubiquitination assays following widely used protocol. We believe that all the blots presented show that SDCBP inhibited FBXO22-mediated BACH1 polyubiquitination in the MDA-MB-231 cell line. As suggested by the reviewer, adding control lane for BACH1 in Fig 3F in which UBCH5a and/or FBXO22 IC would be strengthened this manuscript but would not change our conclusion and also made experimental design too complicated. In addition, re-doing this experiment, we would need to buy the recombinant proteins again, which would be costly and burdensome. Please understand this situation.

About for Figure 4: The reviewer pointed out that controls (like those noted above) are lacking in panels Fig 4D, E, F in the first-round review (In this revised version, Fig 4C and 4D were moved to EV5C and EV5E, respectively, and Fig 4F was moved to Fig. 4E). These Figures investigated the effect of SDCBP on the interaction between BACH1 and FBXO22 using IP assays, or the SKP1-CUL1-FBXO22 complex formation using pull-down assays. We believe that all blots, including control, showed that SDCBP disrupted the SKP1-CUL1-FBXO22 complex formation. We already moved Figure 4C and 4D to the EV Figures and toned down the manuscript as suggested.

In addition, Reviewer concerned regarding Controls without MG132 should be included in panels Fig4 G, H, I. As we responded to the first-round review, researchers typically detect the interaction between E3s and a substrate, and the ubiquitination of a substrates in the presence of MG-132. Previous studies have detected the interaction between SCF-FBXO22 and the substrate for proteasomal degradation, or SCF-FBXO22-mediated ubiquitination of a substrate in the presence of MG-132. As we showed the first-round review, we performed this experiment again to detect the SCF-

FBXO22 BACH1 complex without MG-132. However, we were not able to detect the SCF-FBXO22 BACH1 complex. We believe that these additional data are not necessary to reach any conclusions.

About for Figure 5 and 6, We edited and revised reviewer's concerns regarding these figures in the first-round review.

We understand that reviewer #3 would like to strengthen the data by requesting more control data. Thank you very much for your valuable comments. As suggested by the reviewer, we also re-organized by moving indicated data to EV Figure and reduce the tone of manuscript.

Dear Dr Lee,

Thank you for submitting the revised version of your manuscript. I have now evaluated your amended manuscript and concluded that the remaining minor concerns have been sufficiently addressed.

I am thus pleased to inform you that your manuscript has been accepted for publication in the EMBO Journal.

Related, I would like to hereby ask your consent on keeping the referee figures included in this file.

On a different note, I would like to alert you that EMBO Press offers a format for a video-synopsis of work published with us, which essentially is a short, author-generated film explaining the core findings in hand drawings, and, as we believe, can be very useful to increase visibility of the work. Please see the following link for representative examples and their integration into the article web page:

<https://www.embopress.org/doi/full/10.15252/emj.2019103932>

Best regards,

Daniel Klimmeck

Daniel Klimmeck, PhD
Senior Editor
The EMBO Journal
EMBO
Postfach 1022-40
Meyerhofstrasse 1
D-69117 Heidelberg
contact@embojournal.org
